# Cost-precision trade-off relation determines the optimal morphogen gradient for accurate biological pattern formation

Yonghyun Song, Changbong Hyeon*

Korea Institute for Advanced Study, Seoul, Republic of Korea

**Abstract** Spatial boundaries formed during animal development originate from the pre-patterning of tissues by signaling molecules, called morphogens. The accuracy of boundary location is limited by the fluctuations of morphogen concentration that thresholds the expression level of target gene. Producing more morphogen molecules, which gives rise to smaller relative fluctuations, would better serve to shape more precise target boundaries; however, it incurs more thermodynamic cost. In the classical diffusion-depletion model of morphogen profile formation, the morphogen molecules synthesized from a local source display an exponentially decaying concentration profile with a characteristic length $\lambda$. Our theory suggests that in order to attain a precise profile with the minimal cost, $\lambda$ should be roughly half the distance to the target boundary position from the source. Remarkably, we find that the profiles of morphogens that pattern the *Drosophila* embryo and wing imaginal disk are formed with nearly optimal $\lambda$. Our finding underscores the cost-effectiveness of precise morphogen profile formation in *Drosophila* development.

*For correspondence:
hyeoncb@kias.re.kr

**Competing interests:** The authors declare that no competing interests exist.

## Introduction

Complex spatial structures are shaped across the entire body from an ensemble of initially identical cells during animal development. The emergence of spatial structures is generally linked to the pre-patterning of tissues with biomolecules, called morphogens, that instruct cells to acquire distinct cell fates in a concentration-dependent manner (*Turing, 1990*; *Wolpert, 1969*; *Crick, 1970*). At the molecular level, the positional information encoded in the local morphogen concentration is translated by the cells into the expression of specific genes associated with cell-fate determination. In order to generate reproducible spatial organization, the concentration profile of morphogens must be stable against the noisy background signals inherent to the cellular environment.

Along with quantitative measurements of morphogen gradients (*Gregor et al., 2007*; *Kicheva et al., 2007*; *Bollenbach et al., 2008*; *Kanodia et al., 2009*; *He et al., 2010*; *Zagorski et al., 2017*; *Petkova et al., 2019*), a number of theoretical studies have been devoted to understanding the precision and speed by which morphogen gradients are formed and interpreted (*Tostevin et al., 2007*; *Emberly, 2008*; *Erdmann et al., 2009*; *Berezhkovskii et al., 2010*; *Saunders and Howard, 2009*; *Morishita and Iwasa, 2009*; *Kolomeisky, 2011*; *Tkačik et al., 2015*; *Lo et al., 2015*; *Desponds et al., 2020*; *Fancher and Mugler, 2020*); however, generation of morphogen gradient, which breaks the spatial symmetry, incurs thermodynamic cost, and the relation of this cost with the precision and speed of morphogen gradient formation has rarely been addressed except for a few cases (*Emberly, 2008*). Creating and maintaining morphogen gradients require an influx of energy (*Falasco et al., 2018*), which is a limited resource for biological systems (*Ilker and Hinczewski, 2019*), particularly at the stage of embryonic development (*Rodenfels et al., 2019*;

*Rodenfels et al., 2020*; *Song et al., 2019*). In the present work, we study the trade-off between the cost and precision of the morphogen profile formation in the framework of the reaction-diffusion model of localized synthesis, diffusion, and depletion (SDD model) (*Tostevin et al., 2007*; *Gregor et al., 2007*; *Kicheva et al., 2007*; *Bollenbach et al., 2008*; *Shvartsman and Baker, 2012*; *Teimouri and Kolomeisky, 2014*).

The precision associated with morphogen gradients are perhaps best exemplified by the patterning of the anterior-posterior axis of the *Drosophila* embryo through the Bicoid (Bcd) gradient. In the 2-hr post-fertilization of fruit fly embryogenesis, a uniform field of ~6000 nuclei characterizes the periphery of the shared cytoplasm (*Foe and Alberts, 1983*). Bcd, a transcription factor, is produced from the anterior-end of the embryo by maternally deposited *bcd* mRNAs. Its subsequent diffusion and degradation engender an exponentially decaying profile of Bcd concentration (*Driever and Nüsslein-Volhard, 1988a*; *Driever and Nüsslein-Volhard, 1988b*), which is translated into the anterior expression of the target gene, *hunchback* (*hb*) (*Struhl et al., 1989*). Nuclei at around the middle section of the embryo can infer their relative spatial positions by detecting the concentration of Bcd with an exquisite precision of ~ a single nucleus width (*Gregor et al., 2007*).

Other quantitatively characterized morphogens include Wingless (Wg), Hedgehog (Hh), and Decapentaplegic (Dpp), which pattern the dorsal-ventral (DV) and anterior-posterior (AP) axes of the wing imaginal disk in the fly larvae. Wg, a member of the Wnt signaling pathway, spreads through the wing disk from a narrow band of cells at the DV boundary (*Zecca et al., 1996*; *Neumann and Cohen, 1997*). The Wg concentration profile leads to the differential activation of multiple target genes, and, in particular, induces the sharp expression boundary of *senseless* (*sens*) a few cells away from the DV boundary (*Jafar-Nejad et al., 2006*; *Bakker et al., 2020*). Similarly, Hh patterns the AP axis by spreading from the posterior to the anterior side of the wing disk, affecting the expression of multiple genes. Its downstream transcriptional regulation gives rise to a strip of 8–10 *dpp* expressing cells that form the AP boundary (*Tabata and Kornberg, 1994*; *Strigini and Cohen, 1997*). Dpp, which spreads out from the localized production at the AP boundary, further patterns the AP axis. Major changes in the expression level of Dpp target genes, such as *spalt-major* (*salm*), occur at ~50% of the total length of the domain patterned by Dpp (*Nellen et al., 1996*; *Entchev et al., 2000*). Overall, *Drosophila* wing structures with the spatial precision of a single-cell width emerge from the coordinated actions of multiple patterning events (*Abouchar et al., 2014*).

The positional information, encoded in the concentration profile $\rho(x)$ (red line in *Figure 1*), is decoded to yield the target gene expression profile, $g(\rho(x))$ (blue line in *Figure 1*). The morphogen profile, $\rho(x)$, and the corresponding gene expression level, $g(\rho(x))$, together specify the cell fate in a morphogen concentration-dependent manner. In what follows, we will motivate a quantitative expression for the positional error associated with the 'boundary' positions, where the target gene expression profile displays a sharp change.

We begin by defining a target boundary, $x = x_b$, where the morphogen concentration is at its critical threshold value $\rho_b \equiv \rho(x_b)$. The nuclei exposed to morphogen concentrations higher than $\rho_b$ would adopt an anterior cell fate. Prior to measuring the local morphogen concentration, each nucleus has no information regarding its own position (*Figure 1*). After measuring $\hat{\rho}_b$, the nucleus can estimate the location of target boundary, such that $\hat{x}_b = \hat{x}_b(\hat{\rho}_b)$. For instance, the Bcd concentration at each nucleus at position $x$ ($\rho(x)$) can be detected in terms of the frequency of Bcd binding to regulatory sequences in DNA. Higher local concentration of Bcd leads to a more frequent expression of the target gene, resulting in the position-dependent expression profile of *hb* ($g(\rho(x))$) (*Gregor et al., 2007*). The error (variance) associated with the measured concentration at $x = x_b$ with respect to its true value $\rho_b$ is given by $\sigma_\rho^2(x_b)\left[\equiv \langle (\hat{\rho}_b - \rho_b)^2 \rangle = (1/N)\sum_{i=1}^{N}(\hat{\rho}_{b,i} - \rho_b)^2\right]$. In other words, $\sigma_\rho^2(x_b)$ represents the inherent variability of the morphogen profile, which can be determined experimentally by performing multiple measurements, $\hat{\rho}_{b,i}$ ($i = 1, 2, \cdots N$). For the nucleus to make a single measurement, $\hat{\rho}_{b,i}$, it would integrate the morphogen-induced signal for a short time interval during which the local morphogen concentration remains effectively constant. Then, the Taylor expansion of the concentration at the target boundary, $\rho(\hat{x}_b) \approx \rho(x_b) + (\partial\rho(\hat{x}_b)/\partial\hat{x}_b)|_{\hat{x}_b=x_b}(\hat{x}_b - x_b) + \cdots$, allows one to relate the error in measured concentration with the positional error in estimating the target boundary $\sigma_x^2(x_b)[\equiv \langle (\hat{x}_b - x_b)^2 \rangle = (1/N)\sum_{i=1}^{N}(\hat{x}_{b,i} - x_b)^2]$ as follows:

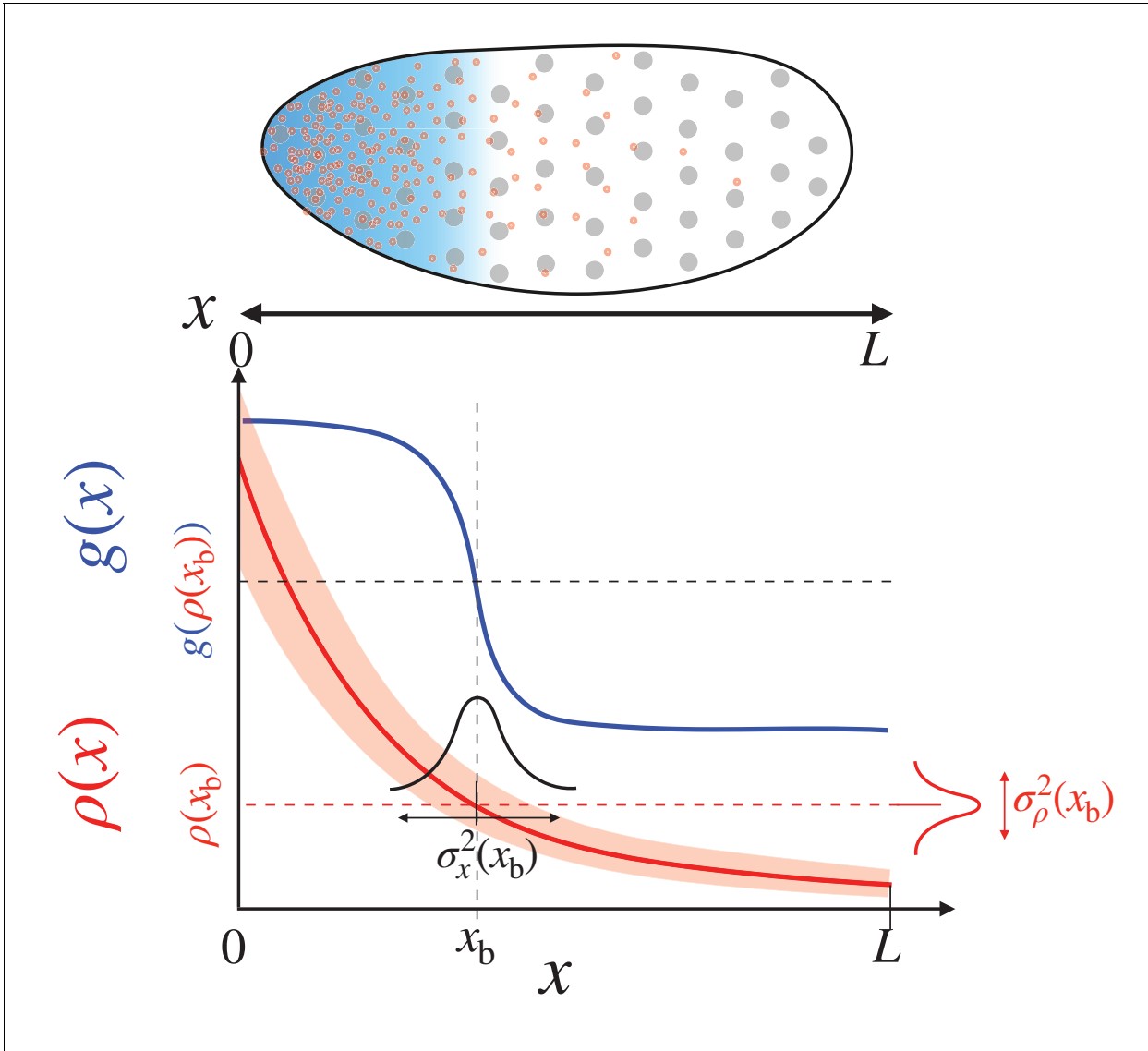

**Figure 1.** Positional information transfer by the morphogen gradient. (Top) The specification of the anterior region of the fruit fly embryo. The uniformly distributed nuclei (gray circles) are subjected to different concentrations of the morphogen (red dots) in the local environment, which leads to the anterior expression of the target gene (blue shade). (Bottom) The red and blue lines respectively depict the morphogen profile, $\rho(x)$, and the target gene expression, $g(x)$, which together specify cell fate. The squared positional error at the boundary $x_b$, $\sigma_x^2(x_b)$, is defined as the product between the variance of the morphogen concentration, $\sigma_\rho^2(x_b)$, and the squared inverse slope of the morphogen profile, $(\partial_x \rho(x))^{-2}_{x=x_b}$.

$$\sigma_x^2(x_b) \simeq \left( \frac{\partial \rho(\hat{x}_b)}{\partial \hat{x}_b} \right)^{-2}_{\hat{x}_b = x_b} \sigma_\rho^2(x_b), \tag{1}$$

such that the morphogen profiles exhibiting a large gradient and small fluctuations give rise to small positional errors.

Of fundamental importance is to address how naturally occurring morphogen profiles, tasked with the transfer of positional information, are formed under limited amount of resources. In an earlier work, Emberly considered the total morphogen content at steady state as a proxy for the cost, and evaluated the 'cost-effectiveness' of the exponentially decaying Bcd profile with a characteristic decay length λ (***Emberly, 2008***). Remarkably, it was shown that for a given positional error, the Bcd profile is shaped with a nearly cost-minimizing λ (***Emberly, 2008***). Here, we expand this argument

with an in-depth treatment of the dynamics of morphogen profile formation and the thermodynamic cost involving the formation and maintenance of precise morphogen profiles.

## Results

The cost of transferring the positional information should include the cellular resources used to generate the steady state profile prior to the measurement, as well as the resources to maintain the profile during the measurement. Theoretically, cells may control the act of measurement by modulating the availability of morphogen-sensing receptors, or by tuning the overall transcription rate of the target gene. In this context, we consider two limiting scenarios. (i) Point measurement, in which the morphogen profile is measured instantaneously; (ii) Space-time-averaged measurement, in which the morphogen profile is measured over a finite space and for a time interval $T$. In the former, the associated cost is defined as the amount of morphogen molecules produced while the profile approaches to the steady state. In the latter, we assume a space-time-averaged measurement carried out for a long time duration. Then, the morphogen produced over the measurement can be approximated as the total cost required to create and maintain the morphogen gradient. For both of the limiting scenarios, we will show that the cost of generating morphogen profiles and the precision of the profiles are counterbalanced. We consider a *morphogen production-independent quantity* by taking the product of total cost and the positional error to quantify the trade-off between the cost and precision of the morphogen profile, and show that the trade-off product can be minimized when the morphogen gradient's characteristic length, λ, is properly selected. We evaluate the cost-effectiveness of morphogen profiles patterning the fruit fly embryo and wing disk, which have been quantitatively characterized in a number of studies (*Gregor et al., 2007*; *Kicheva et al., 2007*; *Bollenbach et al., 2008*; *Nahmad and Stathopoulos, 2009*; *Drocco et al., 2011*; *Wartlick et al., 2011*; *Perry et al., 2012*; *Zhou et al., 2012*; *Chaudhary et al., 2019*; *Petkova et al., 2019*; *Bakker et al., 2020*).

### Point measurement

We begin by defining the dynamics of morphogen profile formation in a system composed of a one-dimensional (1D) array of cells in the domain $0 \leq x \leq L$, where $L$ is the system size (see *Figures 1* and *2(a)*). The term 'cell' refers to the spatial grid in which to define the local positional error and the cost. We denote the amount of morphogen in the cell of the volume ($v_{cell}$) and length ($l_{cell}$) at the interval between $x$ and $x + l_{cell}$ by $\rho(x,t)$. Unless otherwise specified, ρ refers to the ensemble averaged morphogen concentration. At the left boundary (see *Figure 2a*), the morphogen is injected at a constant flux of $j_{in}[\text{conc} \times l_{cell}/\text{time}]$. At all positions, the morphogen molecules are depleted with rate $k_d$ [time$^{-1}$], while spreading across the cells with the diffusivity $D$ (*Figure 2a*). For $L/l_{cell} \gg 1$ (i.e. at the continuum limit), the spatiotemporal dynamics of the morphogen is described using the reaction-diffusion equation

$$\partial_t \rho(x,t) = D \partial_x^2 \rho(x,t) - k_d \rho(x,t), \tag{2}$$

with boundary conditions $-D\partial_x \rho(x,t)|_{x=0} = j_{in}$ and $D\partial_x \rho(x,t)|_{x=L} = 0$. Then, the concentration profile at steady state is obtained as

$$\rho_{ss}(x) = \frac{j_{in}}{\sqrt{Dk_d}} \frac{\cosh\left(\frac{L-x}{\lambda}\right)}{\sinh\left(\frac{L}{\lambda}\right)} \approx \frac{j_{in}}{\sqrt{Dk_d}} e^{-x/\lambda}, \tag{3}$$

where $\lambda = (\sqrt{D/k_d})$ is the characteristic decay length determined by the diffusivity of morphogen ($D$) and the depletion rate ($k_d$). For large system size ($L \gg \lambda$), $\rho_{ss}(x)$ is simply an exponentially decaying profile with characteristic length λ. In what follows, we will define (i) the *cost*, and (ii) the *precision* associated with this reaction-diffusion model.

(i) With a function $R(x,t) \equiv (\rho_{ss}(x) - \rho(x,t))/(\rho_{ss}(x) - \rho(x,0))$, which captures the evolution of the morphogen profile from $\rho(x,t=0) = 0$ to the steady state value $\rho_{ss}(x)$ (*Equation 3*), the mean local accumulation time at $x$ that characterizes the average time scale for establishing the steady state profile is calculated as *Berezhkovskii et al., 2010*,

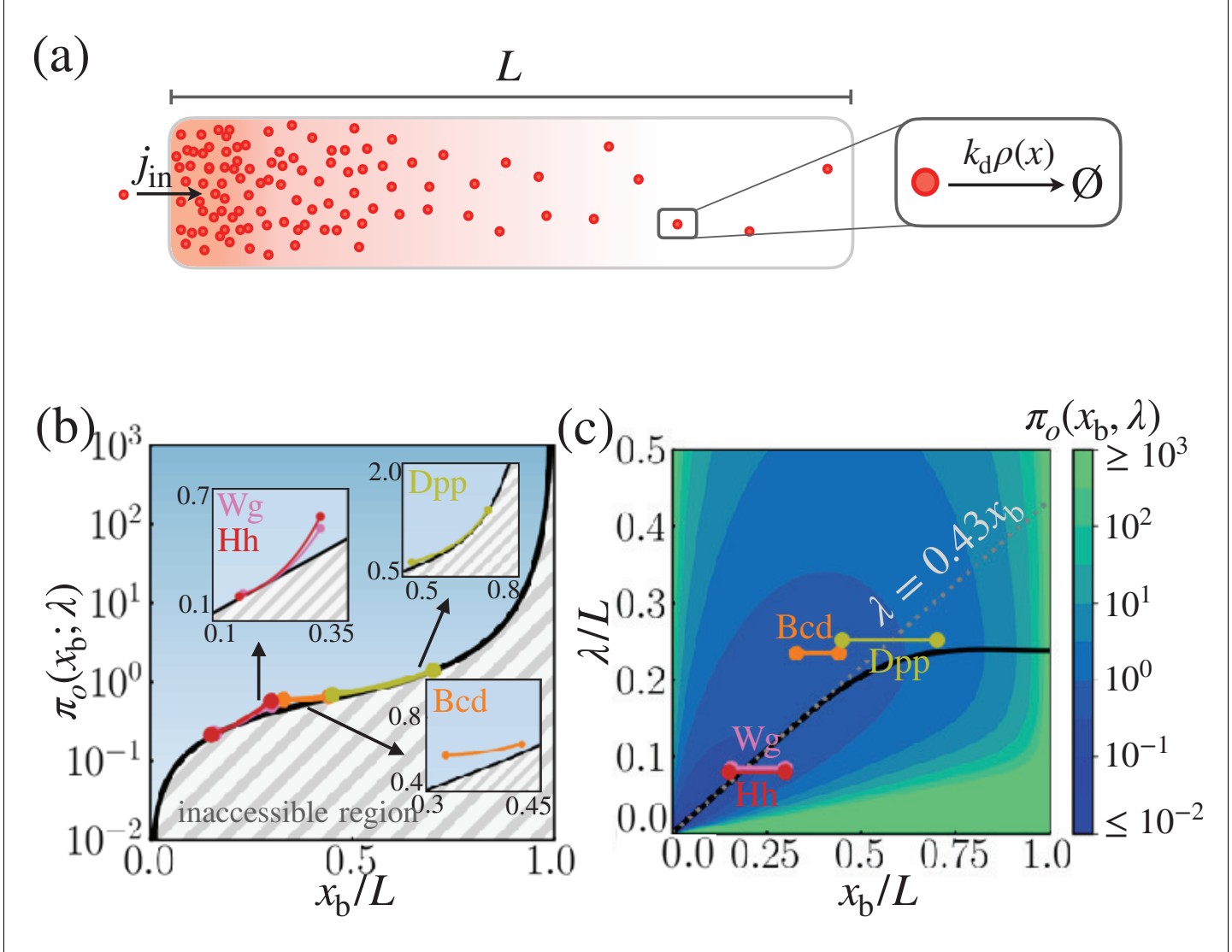

**Figure 2.** Cost-precision trade-off associated with the point measurement. (a) Schematic of the model. (b) The position-dependent lower bound of the trade-off product $\pi_{o,\min}(x_b)$, obtained numerically. The gray hashed area represent the inaccessible regions. The trade-off product of the morphogen profiles of Bcd, Wg, Hh, and Dpp are shown in the respective insets. (c) The black line denotes the optimal characteristic decay length ($\lambda_{\min}$) with respect to the position $x_b/L$. The color scale indicates the trade-off product $\pi_o(x_b)$ computed for each pair of $\lambda/L$ and $x_b/L$ values. The grea dotted line depicts the linear approximation of $\lambda_{\min}$ at large $L$. In (b) and (c), the depicted trade-off product is normalized by $\alpha_o L/l_{\text{cell}}$. The parameters for the naturally occurring morphogen profiles are further described in Appendix 3, Length scales of Bcd, Wg, Hh, and Dpp, and *Appendix 3—table 1*. The online version of this article includes the following figure supplement(s) for figure 2:

**Figure supplement 1.** Length scales of naturally occurring morphogen profiles.

$$\tau(x) = \int_0^\infty R(x,t)dt = k_d^{-1} f(x;\lambda), \tag{4}$$

where $f(x;\lambda) \equiv \frac{1}{2}\left[1 + \frac{L}{\lambda}\coth\left(\frac{L}{\lambda}\right) - \frac{L-x}{\lambda}\tanh\left(\frac{L-x}{\lambda}\right)\right]$ is a dimensionless quantity determined by $x$ and $\lambda$. Then, $v_{\text{cell}}\dot{j}_{\text{in}}/l_{\text{cell}} \times \tau(x)$ quantifies the total number of morphogens produced over the time scale of $\tau(x)$. The thermodynamic cost of producing the morphogens, which is effectively the driving force of pattern formation, is expected to be proportional to this number (*Lynch and Marinov, 2015*), such that

$$C(x) = \alpha_o \left(\frac{v_{\mathrm{cell}}j_{\mathrm{in}}}{l_{\mathrm{cell}}k_{\mathrm{d}}}\right) f(x;\lambda) \approx \alpha_o \left(\frac{v_{\mathrm{cell}}j_{\mathrm{in}}}{l_{\mathrm{cell}}k_{\mathrm{d}}}\right) \frac{1}{2}\left(1 + \frac{x}{\lambda}\right), \tag{5}$$

where $C(x)$ increases linearly with $x$ for large system size ($L \gg \lambda$) (*Berezhkovskii et al., 2010*).

The proportionality constant, $\alpha_o$, amounts to the thermodynamic cost of synthesizing and degrading a single morphogen molecule, which can be quantified by the number of ATPs hydrolyzed in the process. For instance, the thermodynamic cost required to translate and degrade a single Bcd protein composed of 494 amino acids is on the order of $\alpha_o \approx 494 \times 4 \approx 2 \times 10^3$ ATPs, where we assume that each peptide-bond formation requires 4 ATPs (*Lynch and Marinov, 2015*). In the first ~2 hr of *Drosophila* embryogenesis when the anterior-posterior axis patterning takes place, ~$5 \times 10^8$ Bcd molecules are produced (*Drocco et al., 2012*). Thus, the cost of generating the Bcd profile is on the order of $10^{12}$ ATPs, which is a small fraction of the total energy budget of *Drosophila* embryogenesis (~$7 \times 10^{16}$ ATPs) (*Song et al., 2019*). However, the energy budget of developing systems remains largely uncharacterized, and there is certainly a lack of understanding on how much of the cost can be attributed to 'housekeeping processes', as opposed to the cost of generating new spatial structures (*Rodenfels et al., 2019*; *Song et al., 2019*). The generation of each morphogen profile is an indispensable developmental process, although its thermodynamic cost is relatively small compared to the total energy budget.

(ii) The local measure of precision at position $x$ can be quantified by the squared relative error in the positional measurement of morphogen profile (*Equation 1*):

$$\begin{aligned} \epsilon^2(x) &\equiv \frac{\sigma_x^2(x)}{x^2} = \left(\frac{\partial \log \rho_{\mathrm{ss}}(x)}{\partial \log x}\right)^{-2} \frac{\sigma_\rho^2(x)}{\langle \hat{\rho}(x)\rangle^2} \\ &= \left(\frac{\lambda^3}{x^2 l_{\mathrm{cell}}}\right)\left(\frac{v_{\mathrm{cell}}j_{\mathrm{in}}}{l_{\mathrm{cell}}k_{\mathrm{d}}}\right)^{-1} \frac{\sinh\left(\frac{L}{\lambda}\right)\cosh\left(\frac{L-x}{\lambda}\right)}{\left[\sinh\left(\frac{L-x}{\lambda}\right)\right]^2} \\ &\approx \left(\frac{\lambda^3}{x^2 l_{\mathrm{cell}}}\right)\left(\frac{v_{\mathrm{cell}}j_{\mathrm{in}}}{l_{\mathrm{cell}}k_{\mathrm{d}}}\right)^{-1} e^{x/\lambda}, \end{aligned} \tag{6}$$

where $\hat{\rho}(x)$ represents the morphogen concentration estimated at $x$ from a measurement. For large system size ($L \gg \lambda$), $\epsilon^2(x) \propto (\lambda^3/x^2)e^{x/\lambda}$. Note that the multiple measurements of $\hat{\rho}(x)$ yield the mean concentration profile $\langle \hat{\rho}(x)\rangle = (1/N)\sum_{i=1}^{N}\hat{\rho}_i(x)$, which is equivalent to $\rho_{\mathrm{ss}}(x)$ at steady states given in *Equation 3*. We use the fact that the probability distribution of the steady state concentration profile obeys the Poisson statistics (*Heuett and Qian, 2006*) (i.e. $v_{\mathrm{cell}}^2\sigma_\rho^2(x) = v_{\mathrm{cell}}\langle \hat{\rho}(x)\rangle = v_{\mathrm{cell}}\rho_{\mathrm{ss}}(x)$), since all the reactions in the system are of zeroth or first order. Although the present study mainly focuses on the mathematical form of the precision, $\epsilon^2(x)$ is also bounded by physical constraints such as the size of the morphogen sensing machineries that bind and measure the local concentration (*Tostevin et al., 2007*). Experimentally determined values of the precision range from $\epsilon^2(x_{\mathrm{b}}) \approx 7 \times 10^{-4}$ for Bcd to $\epsilon^2(x_{\mathrm{b}}) \approx 1.5 \times 10^{-2}$ for Dpp, where $x_{\mathrm{b}} \approx (0.4 - 0.5)L$ for both systems (*Gregor et al., 2007*; *Bollenbach et al., 2008*).

There is a trade-off between $C(x)$ and $\epsilon^2(x)$. If $n_{\mathrm{m}}$ is the number of morphogen molecules produced for a certain time duration then $C(x) \propto n_{\mathrm{m}}$ and $\sigma_\rho^2(x)/\langle \hat{\rho}(x)\rangle^2 \propto 1/n_{\mathrm{m}}$, the latter of which simply arises from the central limit theorem, or can be rationalized based on the Berg-Purcell result ($\delta\hat{\rho}/\hat{\rho} \sim 1/\sqrt{Da\hat{\rho}\tau}$) (*Berg and Purcell, 1977*) where $a$ is the radius of the volume in which a receptor detects the morphogen and $\tau$ is the detection time. Increasing the overall morphogen content reduces the morphogen profile's positional error at the expense of a larger thermodynamic cost. In *Equations 5 and 6*, $n_{\mathrm{m}}$ corresponds to $(v_{\mathrm{cell}}j_{\mathrm{in}}/l_{\mathrm{cell}}k_{\mathrm{d}})$ (the morphogen molecules produced for the time duration $k_{\mathrm{d}}^{-1}$). The $n_m$-*independent trade-off* between the cost of generating the morphogen profile and the squared relative error in the position of morphogen profile can be quantified by taking the product of the two quantities,

$$\pi_o(x;\lambda) \equiv C(x;\lambda) \times \epsilon^2(x;\lambda) \geq \pi_{o,\mathrm{min}}(x;\lambda), \tag{7}$$

where the $\lambda$ dependence of the trade-off product is made explicit for the discussion that follows.

We are mainly concerned with the precision at the boundary position, $x = x_\mathrm{b}$, where a sharp change in the downstream gene expression, $g(x)$, is observed. The inequality in *Equation 7* constrains the properties of the morphogen profile at $x_\mathrm{b}$, specifying either the minimal cost of generating the morphogen profile for a given positional error or the minimal error in the morphogen profile for a given thermodynamic cost (*Figure 2*). In other words, when the trade-off product $\pi_o(x_\mathrm{b}; \lambda)$ is close to its lower bound $\pi_{o,\min}(x_\mathrm{b}; \lambda)$, the system is cost-effective at generating precise morphogen profiles at $x_\mathrm{b}$. Generally, $\pi_{o,\min}(x_\mathrm{b}; \lambda)$ increases monotonically with $x_\mathrm{b}$, which signifies that boundaries farther away from the origin require the synthesis of more morphogen molecules to achieve comparable positional error (*Figure 2b*). For a fixed system size $L$ and position $x_\mathrm{b}$, the value of the trade-off product is determined solely by the decay length, $\lambda$. By tuning $\lambda$, one can find the value of $\lambda_{\min}$ that minimizes the trade-off product at $x = x_\mathrm{b}$, that is, $\pi_{o,\min}(x_\mathrm{b})$, such that

$$\lambda_{\min}(x_\mathrm{b}) = \arg\min_{\lambda} \pi_o(x_\mathrm{b}; \lambda). \tag{8}$$

The optimal decay length, $\lambda_{\min}(x_\mathrm{b})$, also increases monotonically with the target boundary $x_\mathrm{b}$; for $x_\mathrm{b} < L/2$, $\lambda_{\min}$ is well approximated by the linear relationship $\lambda_{\min}(x_\mathrm{b}) \approx 0.43 x_\mathrm{b}$ (See *Equation 40* for the derivation).

It is of great interest to compare $x_\mathrm{b}$ and $\lambda$ of real morphogen profiles against the optimal $\lambda_{\min}(x_\mathrm{b})$ that one can predict from *Equation 8*. For the morphogen profiles of Bcd, Wg, Hh, and Dpp, we first estimated the possible range of $x_\mathrm{b}$ from the experimentally measured expression profiles of the respective target genes, *hb*, *sens*, *dpp* and *salm* (*Torroja et al., 2004*; *Perry et al., 2012*; *Bakker et al., 2020*). The range of the inferred boundary positions where the target gene expressions undergo relatively sharp changes are shown on the right column of *Figure 2—figure supplement 1* and listed in *Appendix 3—table 1*. The morphogen concentration profiles obtained from experiments are shown on the left column of *Figure 2—figure supplement 1*, where we overlaid the exponential profiles with characteristic lengths $\lambda_\mathrm{Bcd} = 100\,\mu\mathrm{m}$, $\lambda_\mathrm{Wg} = 6\,\mu\mathrm{m}$, $\lambda_\mathrm{Hh} = 8\,\mu\mathrm{m}$, and $\lambda_\mathrm{Dpp} = 20\,\mu\mathrm{m}$ (*Houchmandzadeh et al., 2002*; *Kicheva et al., 2007*; *Wartlick et al., 2011*). *Figure 2b* shows the pairs of $x_\mathrm{b}$ and $\lambda$ normalized by the system size, $L$, which reveals that the $\lambda$'s of naturally occurring morphogen profiles are close to their respective optimal values, $\lambda_{\min}(x_\mathrm{b})$. Taken together, our findings indicate that the concentration profiles of the four morphogens are effectively formed under the condition that the cost-precision trade-off, $\pi_o(x_\mathrm{b}; \lambda)$, is minimized to its lower bound, $\pi_{o,\min}(x_\mathrm{b}; \lambda)$ (*Figure 2b*) (see the Appendix 3, Length scales of Bcd, Wg, Hh, and Dpp, for more details on the relevant parameters of the morphogen profiles).

Our theoretical result must be interpreted with care. For the Bcd profile with the target boundary $x_\mathrm{b} \approx 0.4L$, $\pi_o(x_\mathrm{b}; \lambda_\mathrm{Bcd}) \approx \alpha_o(L/l_\mathrm{cell})0.6$ (*Figure 2b*) where $L/l_\mathrm{cell} \approx 50$, and the experimentally reported precision is $\epsilon^2(x_\mathrm{b}) \approx 7 \times 10^{-4}$ (*Gregor et al., 2007*). Thus, the cost associated with the Bcd profile is $C(x_\mathrm{b}; \lambda_\mathrm{Bcd}) = \pi_o(x_\mathrm{b}; \lambda_\mathrm{Bcd})/\epsilon^2(x_\mathrm{b}; \lambda_\mathrm{Bcd}) \approx 4 \times 10^4 \alpha_o$, or equivalently, the thermodynamic cost of synthesizing and degrading $4 \times 10^4$ Bcd molecules. This is orders of magnitude smaller than the earlier estimate of $\sim 5 \times 10^8$ based on the experimental measurement of total Bcd synthesis in the embryo (*Drocco et al., 2012*). The discrepancy between the two numbers most likely arises because our theory simplifies the dynamics of the nuclear concentration of Bcd in one dimension, whereas the direct measurement of Bcd synthesis accounts for all the molecules in 3D volume including the cytoplasm and the nuclei (*Gregor et al., 2007*; *Drocco et al., 2012*). Additionally, the duration to generate the steady state morphogen profile is longer than the characteristic time, $\tau$. However, assuming that the cytoplasmic and nuclear concentrations are proportional to each other, and that the time for the morphogen profile to reach steady state is proportional to $\tau$, it is possible to adjust the proportionality constant $\alpha_o$ so that $C(x_\mathrm{b})$ represents the overall cost to produce the concentration profile of nuclear Bcd.

## Space-time-averaged measurement

The positional information available from the morphogen profile varies depending on the method by which the cell senses and interprets the local morphogen concentration. Our previous definition of $\epsilon^2(x)$ (*Equation 6*) represents the positional error from a single independent measurement of the morphogen concentration at a specific location in space. However, organisms may further improve the positional information of the morphogen gradient through space-time-averaging.

To incorporate the scenario of space-time-averaging, we assume that the cell at position $x$ uses a sensor with size $a$ to detect the morphogen concentration over the space interval $(x-a, x+a)$ for time $T$ (Figure 3a). Then the space-time-averaged molecular count is written as

$$m_i(x) = \frac{1}{T} \int_0^T dt \int_{x-a}^{x+a} \hat{\rho}_i(y,t)dy,$$

(9)

where $\hat{\rho}_i(y,t)$ represents an estimate of molecular count at position $y$ from an $i$-th independent measurement. Repeated measurements ($N \gg 1$) lead to the mean value of molecular counts

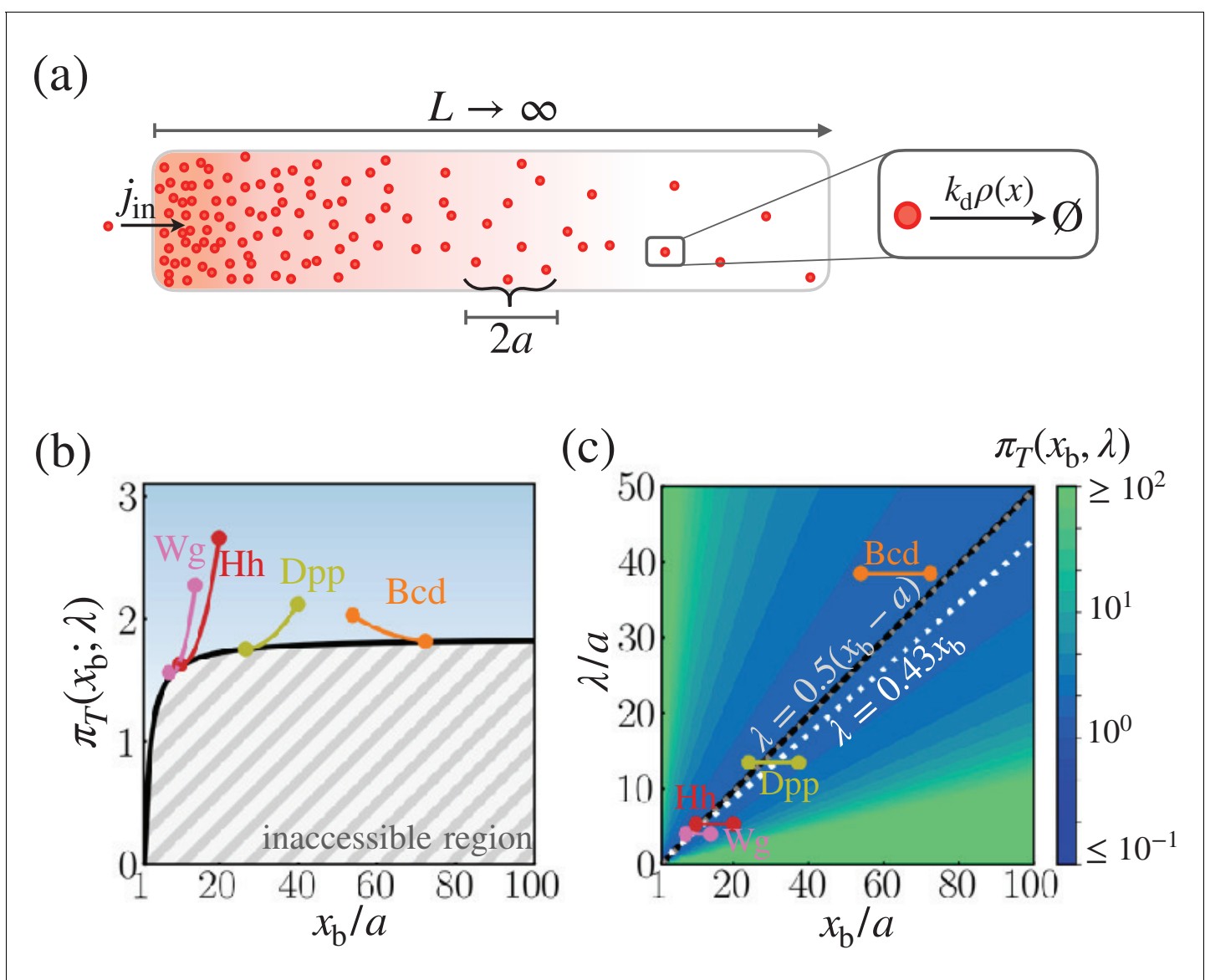

Figure 3. Cost-precision trade-off associated with the space-time-averaged measurement. (a) Schematic of the model. (b) The optimal trade-off product $\pi_T(x_b)$ obtained numerically with respect to the location of the target boundary position normalized by the sensor size ($x_b/a$). The gray hashed area represent the inaccessible regions. Shown are the trade-off products of the morphogens profiles of Bcd, Wg, Hh, and Dpp. (c) The black line denotes the optimal characteristic decay length ($\lambda_{min}$) with respect to the position $x_b/a$. The color scale indicates the trade-off product $\pi_T(x_b)$ computed for each pair of $\lambda/a$ and $x_b/a$ values. The gray dotted line depicts the linear approximation of $\lambda_{min}$. The white dotted line depicts the linear approximation of $\lambda_{min}$ for the point measurement model. In (b) and (c), the depicted trade-off product is normalized by $\alpha_o$. The parameters for the naturally occurring morphogen profiles are further described in the Appendix 3, Length scales of Bcd, Wg, Hh, and Dpp, and Appendix 3—table 1.

$$\langle m(x)\rangle = \frac{1}{N}\sum_{i=1}^{N} m_i(x) = \frac{2j_{\text{in}}}{k_{\text{d}}} e^{-x/\lambda} \sinh(a/\lambda). \tag{10}$$

In this case, $j_{\text{in}}$ and $k_{\text{d}}$ are in the units of $\#/\text{time}$ and $\text{time}^{-1}$, respectively. With a sufficiently long measurement time ($T \gg k_{\text{d}}^{-1}$), the variance of $m(x)$ can be approximated to *Fancher and Mugler, 2020*

$$\sigma_m^2(x) \equiv \frac{1}{N}\sum_{i=1}^{N}(m_i(x) - \langle m(x)\rangle)^2 = \frac{4j_{\text{in}}e^{-x/\lambda}\sinh(a/\lambda)}{Tk_{\text{d}}^2}\left(1 - \frac{(\lambda/a)\cosh(a/\lambda)\sinh(a/\lambda)+1}{2e^{a/\lambda}(\lambda/a)\sinh(a/\lambda)}\right). \tag{11}$$

Analogously to *Equation 6*, the squared relative error (precision) at position $x$ can be quantified as follows,

$$\epsilon_T^2(x) \equiv \left(\frac{\partial \log\langle m(x)\rangle}{\partial \log x}\right)^{-2}\frac{\sigma_m^2(x)}{\langle m(x)\rangle^2} = \frac{(\lambda/x)^2 e^{x/\lambda}}{j_{\text{in}}T}\left(\frac{e^{-a/\lambda}+3e^{3a/\lambda}-4(1+a/\lambda)e^{a/\lambda}}{2(e^{2a/\lambda}-1)^2}\right). \tag{12}$$

The cost of maintaining the morphogen profile at steady states is proportional to $j_{\text{in}}T$, which simply represents the total number of morphogens produced for $T$, such that the total thermodynamic cost for producing morphogens for time $T$ is

$$\mathcal{C}_T = \alpha_o j_{\text{in}}T. \tag{13}$$

In the product between the net amount of morphogen molecules synthesized and the positional error from the averaged signal, the number of morphogens synthesized for time $T$, $j_{\text{in}}T$ cancels off, which yields the following expression of the cost-precision trade-off:

$$\pi_T(x;\lambda) \equiv \mathcal{C}_T \times \epsilon_T^2(x;\lambda) \geq \pi_{T,\text{min}}(x;\lambda). \tag{14}$$

With a given target boundary $x = x_{\text{b}}$ and the sensor size $a$, the value of $\pi_T(x_{\text{b}};\lambda)$ is solely determined by $\lambda$, analogously to $\pi_o(x_{\text{b}};\lambda)$. Thus, the lower bound of $\pi_T(x_{\text{b}};\lambda)$, i.e., $\pi_{T,\text{min}}(x_{\text{b}};\lambda)$, can be determined simply by tuning $\lambda$ to a value that minimizes $\pi_T(x_{\text{b}};\lambda)$, that is, $\lambda_{\text{min}}(x_{\text{b}})$. Both $\pi_{T,\text{min}}(x_{\text{b}};\lambda)$ and $\lambda_{\text{min}}(x_{\text{b}})$ increase monotonically with $x_{\text{b}}$. In particular, $\lambda_{\text{min}}(x_{\text{b}})$ is found in the range of $(x_{\text{b}} - a)/2 \leq \lambda_{\text{min}}(x_{\text{b}}) \leq x_{\text{b}}/2$ (see *Equations 42 and 43* for derivation). The boundary of the inaccessible region in *Figure 3b* constrains $\pi_T(x_{\text{b}};\lambda)$, suggesting that there is a minimal morphogen production for a given positional error. For instance, in order to suppress the positional error down to 10% ($\epsilon_T(x_{\text{b}}) \approx 0.1$) at $x_{\text{b}} = 60a$, the characteristic length must be $\lambda \approx 30a$, which demands that the system synthesize at least ~182 morphogen molecules (*Figure 3b*).

As we have done for $\pi_o$ in the point measurement, we can evaluate $\pi_T$ of the four morphogen profiles at their target boundaries ($x_{\text{b}}$), by using the cell size as a proxy for the sensor size ($a$), and by assuming that the morphogen profile is measured for a sufficiently long time (i.e. $T \gg k_{\text{d}}^{-1}$, the validity of which is further discussed in the Appendix 3, Time scales of Bcd, Wg, Hh, and Dpp). The boundary position and characteristic length associated with each morphogen, normalized by the respective cell size, are shown in *Figure 3c* (see Appendix 3, Length scales of Bcd, Wg, Hh, and Dpp and *Appendix 3—table 1* for more detail). We find that the characteristic decay lengths ($\lambda = \sqrt{D/k_{\text{d}}}$) of all four morphogen profiles are close to their respective $\lambda_{\text{min}}(x_{\text{b}})$ values. Thus, from the space-time-averaged measurement of morphogen profiles, the four morphogen profiles are also formed under the conditions of nearly optimal cost-precision trade-off (*Figure 3b*).

## Discussion

### Comparison of the trade-off products, $\pi_o$ and $\pi_T$

Although the two models of biological pattern formation seem to differ in the definitions of both the precision and the associated cost, the measure of precision in the two models are in fact limiting expressions of each other. The concentration detected in the measurement with space-time-averaging, defined as $m(x)/(2a)$ at the limit of small sensor size ($a/\lambda \ll 1$) (*Equation 10*) is identical to the

one in the point measurement in the limit of $L \gg \lambda$ (*Equation 3*), both yielding $(j_{\mathrm{in}}/\sqrt{Dk_{\mathrm{d}}})e^{-x/\lambda}$. On the one hand, $\epsilon^2(x)$, defined through $\rho_{\mathrm{ss}}(x)$, is experimentally quantifiable through repeated measurements of the morphogen profile from the images of fixed samples (*Kicheva et al., 2007*; *Gregor et al., 2007*; *Bollenbach et al., 2008*). On the other hand, $\epsilon_T^2(x)$, defined through $m(x)$, may represent the precision accessible to cells that integrate the signal over time from multiple receptors positioned across the cell surface. Effectively, $\epsilon^2(x)$ can be interpreted as the short time limit of $\epsilon_T^2(x)$ measured by a single receptor.

The cost associated with the morphogen profile in the case of point measurement, $v_{\mathrm{cell}}j_{\mathrm{in}}l_{\mathrm{cell}}^{-1}\tau(x)$, is a quantity that reflects the number of morphogen molecules required to reach steady state at position $x$. In contrast, the cost of morphogen production with space-time-averaging, $j_{\mathrm{in}}T$, integrated over a time interval $T \gg k_{\mathrm{d}}^{-1}$, is identical for all positions. The latter assumption is required in the derivation of the expression of $\epsilon_T^2(x)$ (see Appendix 1 of *Fancher and Mugler, 2020*). We additionally demand $T \gg \tau(x)$ in order for $j_{\mathrm{in}}T$ to represent the total cost of morphogen profile formation and maintenance. However, $T$ may not necessarily be larger than either $k_{\mathrm{d}}^{-1}$ or $\tau(x)$. For instance, the Bcd profile degrades and stabilizes at time scales of $\sim 1$ hour (*Berezhkovskii et al., 2010*; *Drocco et al., 2012*), but must be measured within 10 min by the nuclei (*Lucas et al., 2018*) (see further discussion in the Appendix 3, Time scales of Bcd, Wg, Hh, and Dpp). The energetic cost of naturally occurring morphogen profiles is likely determined in between the two limiting cases.

For actual biological systems, the cost-precision trade-off involving the pattern formation is presumably at work in between the two scenarios. Thus, of great significance is the finding that $\pi_o(x;\lambda)$ and $\pi_T(x;\lambda)$ for the two limiting models can be effectively minimized by similar $\lambda$ values at a given position: $\lambda_{\min} \approx 0.43x_{\mathrm{b}}$ for the point measurement, and $\lambda_{\min} \approx 0.5x_{\mathrm{b}}$ for the space-time-averaging.

## The entropic cost of forming precise morphogen profiles

Biological pattern formation by morphogen gradients is a process operating out of equilibrium (*Falasco et al., 2018*), in which thermodynamic cost is incurred to generate a morphogen gradient with minimal error for the transfer of positional information against stochastic fluctuations. In the Appendix 1, Reversible reaction-diffusion model of morphogen dynamics, we extend our discussion of cost-precision trade-off on the models that include uni-directional irreversible steps in the synthesis and depletion by considering a more general reaction model with bi-directional reversible kinetics where the forward and reverse rates are well defined at every elementary process (*Appendix 1—figure 1*. A and *Equation 16*). Such a model allows us to quantify the entropy production rate ($\dot{S}_{\mathrm{tot}}$) or the thermodynamic cost for the formation of morphogen gradient and clarifies the physical meaning of $\alpha_o$ (for the relation between $C$ (*Equation 5*) and $\dot{S}_{\mathrm{tot}}$, see Appendix 1 and *Equations 32, 33*). Similar to the point measurement, we derive expressions for the relaxation time to the steady state ($\tau_{\mathrm{rev}}(x)$) and the positional error ($\epsilon_{\mathrm{rev}}^2(x)$). The trade-off among the thermodynamic cost, speed of formation ($\sim \tau_{\mathrm{rev}}^{-1}(x)$), and precision is quantified by the product of the three quantities,

$$\pi_{o,\mathrm{rev}}(x) \equiv \dot{S}_{\mathrm{tot}}\tau_{\mathrm{rev}}(x)\epsilon_{\mathrm{rev}}^2(x) \geq \pi_{o,\mathrm{rev,min}}(x). \tag{15}$$

The lower bound, $\pi_{o,\mathrm{rev,min}}(x_{\mathrm{b}})$, increases monotonically with $x_{\mathrm{b}}$ (*Appendix 1—figure 1*).

With $\dot{S}_{\mathrm{tot}}\tau_{\mathrm{rev}}(x_{\mathrm{b}})$ representing the entropy production during the time scale associated with the morphogen profile formation, *Equation 15* is reminiscent of the thermodynamic uncertainty relation (TUR), a fundamental trade-off relation between the entropy production and the precision of a current-like output observable for dynamical processes generated in nonequilibrium with a universal bound (*Barato and Seifert, 2015*; *Gingrich et al., 2016*; *Hyeon and Hwang, 2017*; *Hwang and Hyeon, 2018*; *Horowitz and Gingrich, 2020*; *Song and Hyeon, 2020*). However, unlike TUR which has a model independent lower bound, the lower bound of *Equation 15* is model-specific and position-dependent. The cost-precision trade-off of pattern formation discussed in this study fundamentally differs from that of TUR (*Pietzonka et al., 2017*; *Horowitz and Gingrich, 2017*; *Dechant and Sasa, 2020*; *Song and Hyeon, 2021*), in that the concentration profile of morphogen is not a current-like observable with an odd-parity with time-reversal; however, still of great significance is the discovery of the underlying principle that the pattern formation is quantitatively bounded by the dissipation. For future work, it would be of great interest to consider employing the morphogen-

induced currents through signaling pathways, such as transcription, as the output observable to which TUR can be directly applied.

## Cost-effectiveness of precise morphogen profile formation

There are multiple possibilities of achieving the same critical threshold concentration $\rho(x_b)$ at the target boundary $x = x_b$ through the combination of morphogen synthesis rate ($j_{in}$), diffusivity ($D$) and

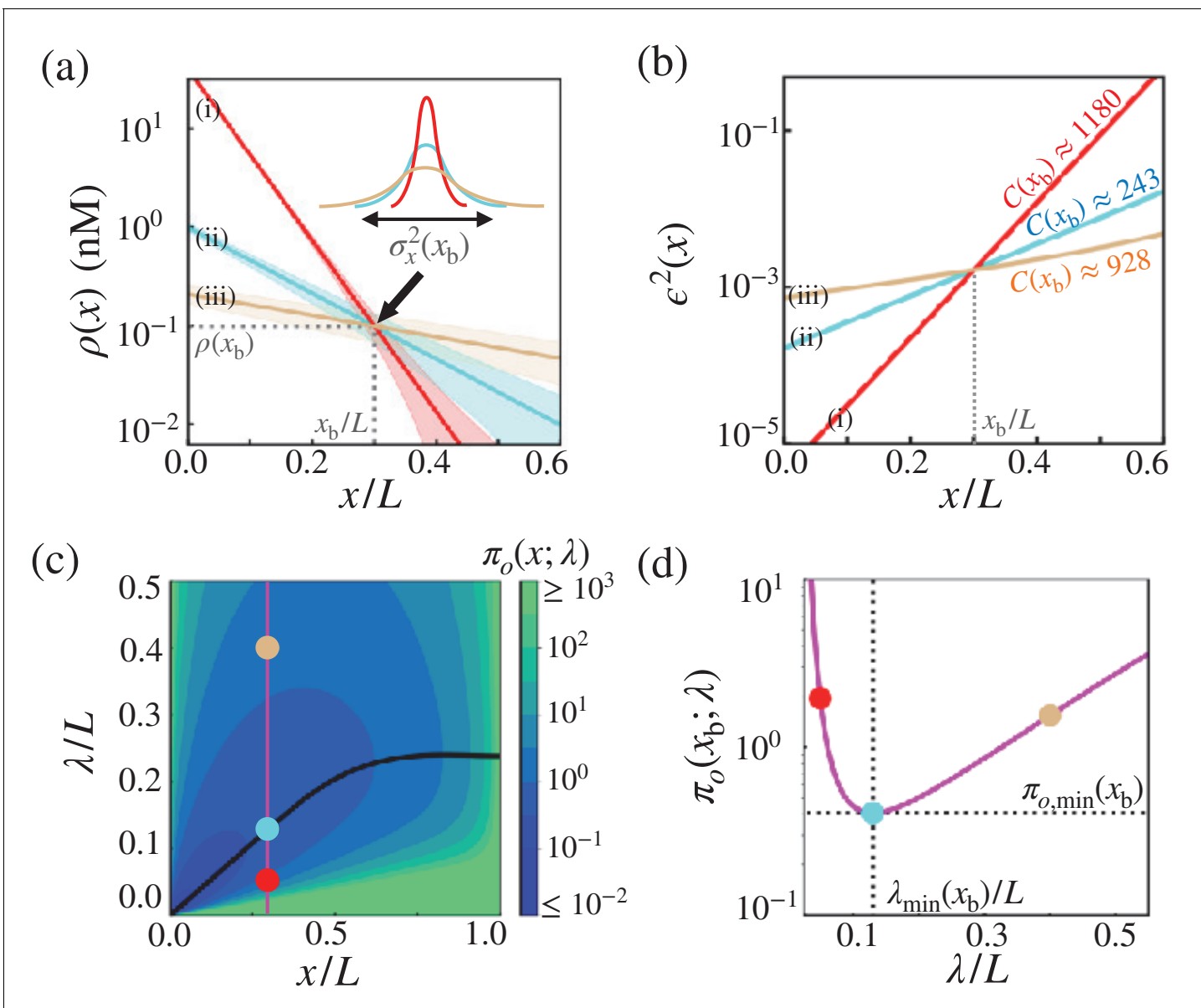

**Figure 4.** Optimal concentration profile of morphogens. (a) Three possible morphogen profiles (i) (red), (ii) (cyan), (iii) (brown) with different $\lambda$'s ($\lambda^{(i)} < \lambda^{(ii)} < \lambda^{(iii)}$), generated with different values of morphogen influxes $j_{in}$ ($j_{in}^{(i)} > j_{in}^{(ii)} > j_{in}^{(iii)}$). The morphogen concentration of the three profiles coincide at $x_b$, giving rise to the same threshold value $\rho(x_b)$ but different positional errors ($\epsilon^2(x_b; \lambda)$). (b) The precision of three possible morphogen profiles (i) (red), (ii) (cyan), (iii) (brown) with different $\lambda$'s ($\lambda^{(i)} < \lambda^{(ii)} < \lambda^{(iii)}$), generated with different values of morphogen influxes $j_{in}$ ($j_{in}^{(i)} > j_{in}^{(iii)} > j_{in}^{(ii)}$). The $\lambda$ values are identical to those with matching colors in (a), but the red and brown curves are generated with different $j_{in}$ values from those in (a). The cost associated with each morphogen profile are shown in units of $\alpha_o L / l_{cell}$. (c) The diagram of the trade-off product associated with the point measurement, $\pi_o(x; \lambda)$, plotted with respect to $x$ and $\lambda$. The black line indicates the optimal decay length, $\lambda_{min}$ at position $x$. Shown on the diagram are the trade-off product $\pi_o$'s for the three cases shown in (a) and (b). (d) The value of $\pi_o$ as a function of $\lambda$ at $x = x_b$. The trade-off product is minimized to $\pi_o = \pi_o^{(ii)} \approx \alpha_o(L/l_{cell})0.4$ with $\pi_o^{(ii)} < \pi_o^{(iii)} < \pi_o^{(i)}$.

depletion rate ($k_\mathrm{d}$) (inset in **Figure 4**). A steeper morphogen profile with larger $j_\mathrm{in}$ and smaller $\lambda(=\sqrt{D/k_\mathrm{d}})$ (red in **Figure 4a**) leads to a more precise boundary but it incurs a higher thermodynamic cost. The opposite case with a morphogen profile with smaller $j_\mathrm{in}$ and larger $\lambda$ gives rise to a less precise boundary with a lower cost (brown in **Figure 4a**). As the main result, we show that $\lambda$ can be tuned to minimize the cost-precision trade-off product, which leads to the formation of cost-effective morphogen profiles. In other words, morphogen profiles with optimal characteristic length $\lambda_\mathrm{min}$ achieve a desired precision when the cost is minimal (**Figure 4b**). Specifically, $\lambda_\mathrm{min} \approx (0.43 - 0.50)x_\mathrm{b}$, which suggests that the target boundary of the exponentially decaying morphogen profile, $c(x) = c_0 e^{-x/\lambda}$, should be formed at $c(x_\mathrm{b}) \approx 0.1c_0$. Remarkably, the $\lambda$'s of naturally occurring morphogen profiles in fruit fly development are close to their respective optimal values, and we confirm that the target boundaries of biological pattern from those profiles are identified at $c(x_\mathrm{b}) \approx 0.1c_0$ (see **Figure 2—figure supplement 1**). These findings lend support to the hypothesis that along with the reduction in the positional error, the thermodynamic cost is one of the key physical constraints in designing the morphogen profiles.

## Concluding remarks

The classical SDD model offers a simple and powerful framework to study basic properties of morphogen dynamics (**Gregor et al., 2007**; **Kicheva et al., 2007**; **Bollenbach et al., 2008**; **Emberly, 2008**; **Berezhkovskii et al., 2010**; **Teimouri and Kolomeisky, 2014**; **Fancher and Mugler, 2020**). The molecular mechanisms underlying the formation of morphogen profiles are, however, much more complex than those discussed here. Even the seemingly simple diffusive spreading of the morphogen can originate from many different mechanisms (**Müller et al., 2013**). In fact, among the four biological examples shown in **Figures 2** and **3**, it has been suggested that Wg and Hh spread over the space through active transport mechanisms rather than through passive diffusion (**Huang and Kornberg, 2015**; **Teimouri and Kolomeisky, 2016**; **Chen et al., 2017**; **Fancher and Mugler, 2020**; **Rosenbauer et al., 2020**). Furthermore, the morphogens considered in the present study induce the expression of multiple target gene expressions, potentially leading to multiple spatial boundaries (**Torroja et al., 2004**; **Hannon et al., 2017**; **Bakker et al., 2020**). While our theory suggests that the cost-precision trade-off can only be optimized at a single target boundary, one can consider an extension of our study in which a weighted average of the precision of many boundary positions is balanced against the total cost of generating the morphogen profile.

Modified trade-off relations in systems with more complex geometries, such as the 1D model with a distributed morphogen source, and the morphogen dynamics on a sphere, can be conceived as well (see Appendix 2). For the latter case, the $\lambda$ values, for the morphogens inducing the endoderm and mesoderm of zebrafish embryos, are found far greater than those leading to the trade-off bound (**Appendix 2—figure 2b**). The large trade-off product values may either simply indicate that the thermodynamic cost is not necessarily a key physical constraint or reflect the presence of other molecular players in a more complex mechanism establishing the zebrafish germ layers. (**Nowak et al., 2011**; **Müller et al., 2012**; **Dubrulle et al., 2015**; **Almuedo-Castillo et al., 2018**).

Generally, multiple morphogen profiles relay combinatorial input signals that determine the expressions of an array of downstream target genes and their subsequent interactions (**Briscoe and Small, 2015**; **Tkačik and Gregor, 2021**). For instance, the anterior patterning of the *Drosophila* embryo depends on the dynamic interpretation of maternal input signals from *bcd*, *nanos*, and *torso* (**Liu et al., 2013**). Nevertheless, each morphogen profile is a fundamental component of biological pattern formation, and its thermodynamic cost must be taken into account in the energy budget of a developing organism. In this light, our theory proposes a quantitative framework to evaluate the cost-precision trade-off of individual morphogen profiles, which allows us to show that the morphogen profiles of fruit fly development are nearly optimal, as opposed to those of the zebrafish embryo. For future work, it would be of great interest to extend our approach to more complex tissue patterning mechanisms, for which key insights are being generated from ongoing studies on the fruit fly, zebrafish, chicken, and mouse (**Dubrulle et al., 2015**; **Oginuma et al., 2017**; **Zagorski et al., 2017**; **Li et al., 2018**; **Petkova et al., 2019**; **Rogers et al., 2020**; **Oginuma et al., 2020**).

## Materials and methods

Methods are described in Appendices 1–3. Appendix 1 derives the expressions associated with the cost-precision trade-off relation for the localized synthesis-diffusion-depletion model of the morphogen dynamics in a 1D array of cells. Appendix 2 derives modified trade-off relations in a 1D model with a distributed morphogen source, and the morphogen dynamics occurring on a sphere. Appendix 3 details how the length and time scales of the naturally occurring morphogen profiles are obtained from literature.

## Acknowledgements

This work was supported by the KIAS individual Grants CG067102 (Y.S.) and CG035003 (C.H.) at Korea Institute for Advanced Study. We thank the Center for Advanced Computation in KIAS for providing computing resources.

## Additional information

### Funding

| Funder | Grant reference number | Author |
|---|---|---|
| Korea Institute for Advanced Study | CG067102 | Yonghyun Song |
| Korea Institute for Advanced Study | CG035003 | Changbong Hyeon |

The funders had no role in study design, data collection and interpretation, or the decision to submit the work for publication.

### Author contributions

Yonghyun Song, Conceptualization, Formal analysis, Validation, Investigation, Visualization, Writing - original draft, Writing - review and editing; Changbong Hyeon, Conceptualization, Formal analysis, Supervision, Funding acquisition, Validation, Investigation, Writing - original draft, Project administration, Writing - review and editing

### Author ORCIDs

Yonghyun Song https://orcid.org/0000-0002-8713-1294
Changbong Hyeon https://orcid.org/0000-0002-4844-7237

### Decision letter and Author response

Decision letter https://doi.org/10.7554/eLife.70034.sa1
Author response https://doi.org/10.7554/eLife.70034.sa2

## Additional files

### Supplementary files

• Transparent reporting form

### Data availability

All data analyzed in this study are from the figures of previously published works, shown in Figure 2-S1 and Appendix 2 Figure 3. The reference associated with each panel is provided in the respective figure legend.

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

## Appendix 1

### Reversible reaction-diffusion model of morphogen dynamics

Here, we provide detailed derivations of the cost-precision trade-off relation for the localized synthesis-diffusion-depletion model of the morphogen dynamics in a 1D array of cells. We begin with the reversible reaction-diffusion process of morphogen profile formation, where the forward and reverse rates are well defined at every elementary step, which enables the calculation of the total entropy production rate ($\dot{S}_{\text{tot}}$). Through the discussion of reversible process, we provide mathematical details on the derivation of the cost-precision trade-off relation for the irreversible reaction-diffusion process of morphogen synthesis and depletion presented in the main text. Next, we derive the conditions that lead to the minimum trade-off product for the irreversible process with point ($\pi_o$) and space-time-averaged ($\pi_T$) measurements.

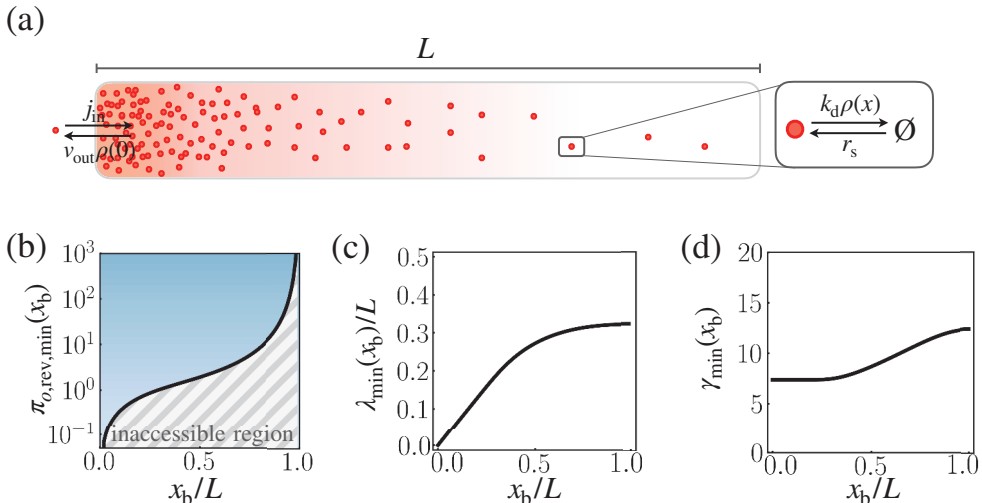

**Appendix 1—figure 1.** Speed-precision-cost trade-off for the reversible morphogen dynamics. (**a**) Schematic of the reversible reaction-diffusion dynamics of morphogen profile formation. (**b**) The solid black line depicts the optimal trade-off product $\pi_{o,\text{rev,min}}(x_{\text{b}})$, obtained numerically by minimizing $\pi_{o,\text{rev}}(x_{\text{b}})$ with respect to $\lambda$ and $\gamma$, with $\beta_{\text{out}} \to \infty$. The trade-off product is normalized by $L/l_{\text{cell}}$. (**c,d**) The optimal characteristic decay length $\lambda_{\text{min}}$ (b) and $\gamma_{\text{min}}$ (c) corresponding to $\pi_{o,\text{rev,min}}(x_{\text{b}})$.

We explore the trade-offs among the speed, precision, and entropy production in the reversible process of morphogen dynamics. Similarly to the irreversible version presented in the main text, the dynamics is defined on a 1D array of cells. We denote the amount of morphogen in the cell of volume ($v_{\text{cell}}$) and size ($l_{\text{cell}}$) at the interval between $x$ and $x + l_{\text{cell}}$ by the concentration, $\rho(x, t)$. At the left boundary (see *Appendix 1—figure 1a*), the morphogen is injected with rate $j_{\text{in}}[\text{conc} \times l_{\text{cell}}/\text{time}]$, and taken out by the corresponding reverse reaction with rate $v_{\text{out}}[l_{\text{cell}}/\text{time}]$. At all positions, the morphogen molecules are depleted with rate $k_{\text{d}}$ and synthesized by the corresponding reverse reaction with rate $r_{\text{s}}$, and spread across space with diffusivity $D$. At the continuum limit ($L \gg l_{\text{cell}}$), the dynamics of the morphogen concentration is described by the partial differential equation

$$\partial_t \rho(x, t) = D \partial_x^2 \rho(x, t) - k_{\text{d}} \rho(x, t) + r_{\text{s}}, \tag{16}$$

with two boundary conditions at $x = 0$ and $x = L$, $-D\partial_x \rho(x,t)|_{x=0} = j_{\text{in}} - v_{\text{out}}\rho(0,t)$ and $D\partial_x \rho(x,t)|_{x=L} = 0$. $\lambda \equiv \sqrt{D/k_{\text{d}}}$ is the characteristic length of the dynamics associated with *Equation 16*. The steady state concentration is expressed as

$$\rho_{\text{ss}}(x) = \underbrace{\frac{r_{\text{s}}}{k_{\text{d}}} \frac{(\gamma - 1)\cosh\left[\frac{L-x}{\lambda}\right]}{\beta_{\text{out}}^{-1}\sinh\left[\frac{L}{\lambda}\right] + \cosh\left[\frac{L}{\lambda}\right]}}_{=\rho_{\text{ss,0}}(x)} + \frac{r_{\text{s}}}{k_{\text{d}}}, \tag{17}$$

where $\rho_{ss,0}(x)$ denotes the position-dependent portion of the concentration profile. We introduce three dimensionless parameters: $\beta_{in} \equiv j_{in}/(\lambda r_s)$, $\beta_{out} \equiv v_{out}/(\lambda k_d)$, and their ratio $\gamma \equiv \beta_{in}/\beta_{out}$ which is related with the thermodynamic cost (see *Equation 32*). We only consider the regime with $\gamma > 1$, where a net positive current of morphogen is supplied at the left boundary. In what follows, we will derive the quantitative expressions of the (i) *speed*, (ii) *precision*, and (iii) *thermodynamic cost* associated with the morphogen profile.

## Speed

The concentration profile varies with time to reach its steady state, $\rho_{ss}(x) \equiv \lim_{t \to \infty} \rho(x,t)$, from the initial profile $\rho(x,0) = 0$. The time evolution of $\rho(x,t)$ can be analyzed by solving the *Equation 16* in Laplace domain:

$$\hat{\rho}(x,s) = \frac{1}{s}\left[\frac{(j_{in}(s+k_d) - r_s v_{out})\cosh\left[(L-x)\sqrt{(s+k_d)/D}\right]}{(s+k_d)\sinh\left[L\sqrt{(s+k_d)/D}\right]\left(D\sqrt{(s+k_d)/D} + v_{out}\coth\left[L\sqrt{(s+k_d)/D}\right]\right)} + \frac{r_s}{(s+k_d)}\right]. \quad (18)$$

The expression of the steady state profile (*Equation 17*) is obtained using

$$\lim_{s \to 0}(s\hat{\rho}(x,s) - \rho(x,0)) = \lim_{s \to 0}\int_0^\infty dt e^{-st}\frac{d\rho(x,t)}{dt} = \int_0^\infty dt\frac{d\rho(x,t)}{dt} = \rho(x,\infty) - \rho(x,0) = \rho_{ss}(x). \quad (19)$$

To characterize the relaxation dynamics to the steady state profile, we define the relaxation function

$$R(x,t) \equiv \frac{\rho_{ss}(x) - \rho(x,t)}{\rho_{ss}(x) - \rho(x,0)} = 1 - \frac{\rho(x,t)}{\rho_{ss}(x)}, \quad (20)$$

which monotonically decays from 1 to 0 at all positions as $t \to \infty$. The associated mean relaxation time at position $x$, $\tau_{rev}(x)$, is given by

$$\tau_{rev}(x) = \int_0^\infty R(x,t)dt. \quad (21)$$

Since the Laplace transform of the relaxation function is $\hat{R}(x,s) = \int_0^\infty e^{-st}R(x,t)dt = s^{-1} - \hat{\rho}(x,s)/\rho_{ss}(x)$, the local accumulation time at position $x$, $\tau_{rev}(x) = \lim_{s \to 0}\hat{R}(x,s)$, is obtained as

$$\tau_{rev}(x) = \mathcal{N}/\mathcal{M} \quad (22)$$

with

$$\begin{aligned}
\mathcal{N} \equiv\ &2\beta_{out}(\coth[L/\lambda])^2 \\
&+ \operatorname{csch}\left[\frac{L}{\lambda}\right]\left\{\left[(\gamma-3) + \beta_{out}(\gamma-1)\frac{L}{\lambda}\right]\cosh\left[\frac{L-x}{\lambda}\right] + 2\beta_{out}^{-1}\sinh\left[\frac{L}{\lambda}\right] - (\gamma-1)\frac{L-x}{\lambda}\sinh\left[\frac{L-x}{\lambda}\right]\right\} \\
&+ \coth\left[\frac{L}{\lambda}\right]\operatorname{csch}\left[\frac{L}{\lambda}\right]\left(\left[\gamma\frac{L}{\lambda} - \left(2\beta_{out} + \frac{L}{\lambda}\right)\right]\cosh\left[\frac{L-x}{\lambda}\right] + 4\sinh\left[\frac{L}{\lambda}\right] - \beta_{out}(\gamma-1)\frac{L-x}{\lambda}\sinh\left[\frac{L-x}{\lambda}\right]\right)
\end{aligned}$$

and

$$\mathcal{M} \equiv 2k_d\left(1 + \beta_{out}\coth\left[\frac{L}{\lambda}\right]\right)\left(\beta_{out}^{-1} + \coth\left[\frac{L}{\lambda}\right] + (\gamma-1)\cosh\left[\frac{L-x}{\lambda}\right]\operatorname{csch}\left[\frac{L}{\lambda}\right]\right).$$

$\tau_{rev}^{-1}(x)$ quantifies the speed at which the morphogen profile is established at position $x$ (*Berezhkovskii et al., 2010*; *Berezhkovskii et al., 2011*). The expression for the characteristic time in the irreversible morphogen dynamics is obtained when $r_s$ and $v_{out}$ are negligibly small. At the limits of $r_s \to 0$ and $v_{out} \to 0$, with the substitutions $\beta_{out} = v_{out}/(\lambda k_d)$ and $\gamma = j_{in}k_d/(v_{out}r_s)$, *Equation 22* is simplified to

$$\tau(x) = \lim_{\substack{r_s \to 0 \\ v_{out} \to 0}} \tau_{rev}(x) = \frac{1}{2k_d}\left(1 + \frac{L}{\lambda}\coth\left[\frac{L}{\lambda}\right] - \frac{L-x}{\lambda}\tanh\left[\frac{L-x}{\lambda}\right]\right). \tag{23}$$

## Precision

The local measure of precision at $x$ is defined as the squared relative positional error (*Equation 1*),

$$\begin{aligned}
\epsilon^2_{rev}(x) &\equiv \frac{\sigma_x^2(x)}{x^2} = \left(\frac{\partial \log \rho_{ss}(x)}{\partial \log x}\right)^{-2}\frac{\sigma_\rho^2(x)}{\rho_{ss}(x)^2} \\
&= \frac{\lambda^2 k_d}{v_{cell}x^2 r_s}\frac{\left(\beta_{out}^{-1} + \coth\left[\frac{L}{\lambda}\right]\right)\left(\beta_{out}^{-1} + \coth\left[\frac{L}{\lambda}\right] + (\gamma-1)\cosh\left[\frac{L-x}{\lambda}\right]\operatorname{csch}\left[\frac{L}{\lambda}\right]\right)\left(\operatorname{csch}\left[\frac{L-x}{\lambda}\right]\sinh\left[\frac{L}{\lambda}\right]\right)^2}{(\gamma-1)^2}.
\end{aligned} \tag{24}$$

The expression for the squared relative error in the irreversible process of morphogen dynamics can be obtained at the limits of $r_s \to 0$ and $v_{out} \to 0$, with the substitutions $\beta_{out} = v_{out}/(\lambda k_d)$ and $\gamma = j_{in}k_d/(v_{out}r_s)$:

$$\epsilon^2(x) = \lim_{\substack{r_s \to 0 \\ v_{out} \to 0}} \epsilon^2_{rev}(x) = \frac{\lambda^3}{l_{cell}x^2}\left(\frac{v_{cell}j_{in}}{l_{cell}k_d}\right)^{-1}\sinh\left[\frac{L}{\lambda}\right]\cosh\left[\frac{L-x}{\lambda}\right]\left(\sinh\left[\frac{L-x}{\lambda}\right]\right)^{-2}. \tag{25}$$

## Thermodynamic cost

The total entropy production rate required to maintain the steady state morphogen profile is given by

$$\dot{S}_{tot} = \frac{1}{l_{cell}}\int_0^L dx\left[\dot{S}_{in}(x) + \dot{S}_{dep}(x) + \dot{S}_{diff}(x)\right], \tag{26}$$

where the three components of the position-dependent entropy production rate are

$$\dot{S}_{in}(x) \equiv 2v_{cell}(j_{in} - v_{out}\rho_{ss}(0))\delta(x)\log\left(\frac{j_{in}}{v_{out}\rho_{ss}(0)}\right), \tag{27}$$

$$\dot{S}_{dep}(x) \equiv v_{cell}(k_d\rho_{ss}(x) - r_s)\log\left(\frac{k_d\rho_{ss}(x)}{r_s}\right), \tag{28}$$

$$\dot{S}_{diff}(x) \equiv v_{cell}D[\partial_x\rho_{ss}(x)][\partial_x\ln\rho_{ss}(x)] = v_{cell}D\frac{[\partial_x\rho_{ss}(x)]^2}{\rho_{ss}(x)}, \tag{29}$$

with the Boltzmann constant set to unity ($k_B = 1$). *Equations 27 and 28* represent the entropy production rates from the net morphogen current flowing in and out of the system, respectively. Similarly, *Equation 29* arises from the diffusive flux (*Falasco et al., 2018*). Briefly, *Equation 29* can be understood as $\lim_{l_{cell} \to 0} Dl_{cell}^{-2}(\rho_{ss}(x+l_{cell}) - \rho_{ss}(x))\ln[\rho_{ss}(x+l_{cell})/\rho_{ss}(x)]$, in which $D\rho_{ss}(x+l_{cell})/l_{cell}^2$ and $D\rho_{ss}(x)/l_{cell}^2$ amount to the forward and reverse currents between adjacent spatial compartments in the discretized spatial representation. It is noteworthy that the asymmetry of concentrations generates a flow of morphogen current between the neighboring cells, and contributes to the entropy production. If the morphogen profile is uniform in space, which occurs at the detailed balance condition ($\dot{S}_{tot} = 0$), the cells cannot infer any spatial information from it.

To obtain a simplified expression of $\dot{S}_{tot}$, we use the following two equalities. First, at steady state, the net injection of the morphogen at the left-boundary must be equal to the net depletion of the morphogen occurring at all positions, so that

$$\int_0^L 2\delta(x)[j_{in} - v_{out}\rho_{ss}(0)]dx = \int_0^L [k_d\rho_{ss}(x) - r_s]dx. \tag{30}$$

Second, $\partial_x \rho_{ss}(x) = \partial_x \rho_{ss,0}(x)$ and $\partial_x^2 \rho_{ss}(x) = \partial_x^2 \rho_{ss,0}(x) = \lambda^{-2} \rho_{ss,0}(x)$, with $\rho_{ss,0}$ defined in *Equation 17*, lead to the following equality:

$$\int_0^L \left[ \rho_{ss,0}(x) \ln \rho_{ss}(x) \right] dx = \lambda^2 \left( \left[ \partial_x \rho_{ss,0}(x) \right] \ln \rho(x) \Big|_0^L - \int_0^L \left[ \frac{[\partial_x \rho_{ss}(x)]^2}{\rho_{ss}(x)} \right] dx \right). \tag{31}$$

Then, $\dot{S}_{tot}$ can be expressed as follows:

$$\begin{aligned}
\dot{S}_{tot} &= \frac{1}{l_{cell}} \int_0^L \left( \dot{S}_{in}(x) + \dot{S}_{dep}(x) + \dot{S}_{diff}(x) \right) dx \\
&= \frac{v_{cell} k_d}{l_{cell}} \int_0^L \left( \rho_{ss,0}(x) \ln \frac{j_{in} k_d}{v_{out} r_s} + \rho_{ss,0}(x) \ln \frac{\rho_{ss}(x)}{\rho_{ss}(0)} + \lambda^2 \frac{[\partial_x \rho_{ss}(x)]^2}{\rho_{ss}(x)} \right) dx \\
&= \frac{v_{cell} k_d}{l_{cell}} \left( \int_0^L \rho_{ss,0}(x) \ln \gamma \, dx - \int_0^L \rho_{ss,0}(x) \ln \rho_{ss}(0) dx + \int_0^L \left[ \rho_{ss,0}(x) \ln \rho_{ss}(x) + \lambda^2 \frac{[\partial_x \rho_{ss}(x)]^2}{\rho_{ss}(x)} \right] dx \right) \\
&= \frac{v_{cell} k_d}{l_{cell}} \left( \int_0^L \rho_{ss,0}(x) \ln \gamma \, dx - \lambda^2 [\partial_x \rho_{ss,0}(x)] \ln \rho_{ss}(0) \Big|_0^L + \lambda^2 [\partial_x \rho_{ss,0}(x)] \ln \rho_{ss}(x) \Big|_0^L \right) \\
&= \frac{v_{cell} k_d}{l_{cell}} \ln \gamma \int_0^L \rho_{ss,0}(x) dx \\
&= \frac{v_{cell} \lambda r_s}{l_{cell}} \frac{(\gamma - 1) \ln \gamma}{\beta_{out}^{-1} + \coth\left[ \frac{L}{\lambda} \right]}.
\end{aligned} \tag{32}$$

We used 30 and 31 to obtain the second and fourth lines of *Equation 32*, respectively. The boundary condition of the system, $\partial_x \rho(x)|_{x=L} = 0$, and the integral of $\rho_{ss,0}(x)$ leads to the last line.

The reversible model motivates a physical interpretation of the proportionality constant $\alpha_o$, which we previously defined in the irreversible model of morphogen dynamics (*Equation 2*) as the number of ATPs hydrolyzed for the synthesis and depletion of a morphogen molecule (*Equation 5*). When $\dot{S}_{tot}$ (*Equation 32*) is normalized by $\ln \gamma$, and $r_s$ and $v_{out}$ are negligibly small, we obtain

$$\lim_{\substack{r_s \to 0 \\ v_{out} \to 0}} \frac{\dot{S}_{tot}}{\ln \gamma} = \frac{v_{cell} j_{in}}{l_{cell}}. \tag{33}$$

Comparison of *Equation 33* with the expression in *Equation 5* enables us to relate $\alpha_o$ with the free energy cost, $T \ln \gamma$. Thus, the thermodynamic cost associated with the morphogen profile is expressed as $C = (\alpha_o v_{cell} j_{in}/l_{cell}) \tau$ (*Equation 5*) where $\tau$ is the time required to generate the profile, and $\alpha_o v_{cell} j_{in}/l_{cell}$ approximates the rate of dissipation, $T \dot{S}_{tot}$.

## Trade-off product

The product of the local accumulation time ($\tau_{rev}(x)$, *Equation 22*), the squared relative positional error ($\epsilon_{rev}^2(x)$, *Equation 24*), and the entropy production rate ($\dot{S}_{tot}$, *Equation 32*) leads to the position-dependent trade-off product $\pi_{o,rev}(x)$,

$$\begin{aligned}
\pi_{o,rev}(x; \lambda, \beta_{out}, \gamma) &= \dot{S}_{tot} \tau_{rev}(x) \epsilon_{rev}^2(x) \\
&= \frac{1}{2} \frac{L}{l_{cell}} \frac{\lambda^3}{Lx^2} \frac{\operatorname{csch}[(L-x)/\lambda] \sinh[L/\lambda] \ln \gamma}{(\gamma - 1)(\beta_{out} \coth[L/\lambda] + 1)} \\
&\quad \times \left\{ (1 - \gamma) \frac{L-x}{\lambda} + \left( \gamma - 3 + \beta_{out}(\gamma - 1) \frac{L}{\lambda} \right) \coth\left[ \frac{L-x}{\lambda} \right] + 4 \cosh\left[ \frac{L}{\lambda} \right] \operatorname{csch}\left[ \frac{L-x}{\lambda} \right] \right. \\
&\quad + \coth\left[ \frac{L}{\lambda} \right] \left( \beta_{out}(1 - \gamma) \frac{L-x}{\lambda} + \left( (\gamma - 1) \frac{L}{\lambda} - 2\beta_{out} \right) \coth\left[ \frac{L-x}{\lambda} \right] \right. \\
&\quad \left. + 2\beta_{out} \cosh\left[ \frac{L}{\lambda} \right] \operatorname{csch}\left[ \frac{L-x}{\lambda} \right] \right) \\
&\quad \left. + 2\beta_{out}^{-1} \operatorname{csch}\left[ \frac{L-x}{\lambda} \right] \sinh\left[ \frac{L}{\lambda} \right] \right\}.
\end{aligned} \tag{34}$$

With fixed $L$, $l_{cell}$, and $x = x_b$, the lower bound of $\pi_{o,rev}(x_b; \lambda, \beta_{out}, \gamma)$, $\pi_{rev,min}(x_b)$, can be obtained

by minimizing $\pi_{o,\text{rev}}(x_b; \lambda, \beta_{\text{out}}, \gamma)$ with respect to $\gamma$, $\beta_{\text{out}}$, and $\lambda$. For a fixed value of $\gamma > 1$, the partial derivative of $\pi_{o,\text{rev}}(x_b; \lambda, \beta_{\text{out}}, \gamma)$ with respect to $\beta_{\text{out}}$ is always smaller than 0,

$$
\begin{aligned}
&\partial_{\beta_{\text{out}}} \pi_{o,\text{rev}}(x_b; \lambda, \beta_{\text{out}}, \gamma) \\
&= -\frac{L}{l_{\text{cell}}} \frac{\lambda^3}{L(x_b)^2} \ln \gamma \left( \text{csch}\left[\frac{L - x_b}{\lambda}\right] \right)^2 \\
&\times \left( \frac{(\sinh[L/\lambda])^2}{\beta_{\text{out}}^2(\gamma - 1)} + \frac{\cosh[(L - x_b)/\lambda]\sinh[L/\lambda](2L/\lambda + \sinh[2L/\lambda])}{4(\beta_{\text{out}}\cosh[L/\lambda] + \sinh[L/\lambda])^2} \right) < 0.
\end{aligned}
\tag{35}
$$

Thus, $\pi_{o,\text{rev}}(x_b)$ is minimized at the limit of large $\beta_{\text{out}}$ (or, equivalently, large $\beta_{\text{in}} = \gamma\beta_{\text{out}}$), with the following expression:

$$
\begin{aligned}
\lim_{\beta_{\text{out}} \to \infty} \pi_{o,\text{rev}}(x_b; \lambda, \beta_{\text{out}}, \gamma) &= \frac{1}{4}\frac{L}{l_{\text{cell}}}\frac{\lambda^3}{L(x_b)^2}\frac{(\text{csch}[(L - x_b)/\lambda])^2 \tanh[L/\lambda] \ln \gamma}{(\gamma - 1)} \\
&\times \left\{ 2 + 2\cosh\left[\frac{2L}{\lambda}\right] - 2\cosh\left[\frac{2L - x_b}{\lambda}\right] - 2\cosh\left[\frac{x_b}{\lambda}\right] \right. \\
&\left. + (\gamma - 1)\left(\frac{x_b}{\lambda}\sinh\left[\frac{2L - x_b}{\lambda}\right] + \left(\frac{2L - x_b}{\lambda}\right)\sinh\left[\frac{x_b}{\lambda}\right]\right) \right\}.
\end{aligned}
\tag{36}
$$

With the characteristic decay length set to $\lambda = x_b$, it can be shown that $\lim_{\substack{x_b \to 0 \\ \beta_{\text{out}} \to \infty}} \pi_{o,\text{rev}}(x_b; x_b, \beta_{\text{out}}, \gamma) = 0$. Since $\pi_{o,\text{rev,min}}(x_b) \leq \pi_{o,\text{rev}}(x_b; x_b, \beta_{\text{out}}, \gamma)$ by the definition of $\pi_{o,\text{rev,min}}(x_b)$, $\pi_{o,\text{rev,min}}(0) = 0$.

For $x_b > 0$, the optimal trade-off product, $\pi_{\text{rev,min}}(x_b)$, and the corresponding values of $\lambda_{\text{min}}(x_b)$ and $\gamma_{\text{min}}(x_b)$, were numerically obtained (*Appendix 1—figure 1*). $\pi_{o,\text{rev,min}}(x_b)$ constrains dynamical properties of the morphogen profile, specifying, for instance, the minimum entropy production rate for a given speed and precision (*Appendix 1—figure 1b*). With a small $\pi_{o,\text{rev}}(x_b; \lambda, \beta_{\text{out}}, \gamma)$ close to $\pi_{o,\text{rev,min}}(x_b)$, the system can rapidly generate precise morphogen profiles at $x_b$ with low thermodynamic cost. In general, $\pi_{o,\text{rev,min}}(x_b)$, $\lambda_{\text{min}}(x_b)$, and $\gamma_{\text{min}}(x_b)$ all increase monotonically with $x_b$ (*Appendix 1—figure 1b,c,d*).

A cautionary remark is in place. A direct one-to-one comparison between the trade-off products of the reversible and irreversible models should be avoided, since $\pi_{o,\text{rev}}$ is minimized at the limits of $v_{\text{out}}, r_s \to \infty$ (see *Equation 35*), while the irreversible model is equivalent to the reversible version at the limits of $v_{\text{out}}, r_s \to 0$.

## Minimum trade-off product of the point measurement for the irreversible reaction-diffusion process

We provide the mathematical details involving the trade-off product of the point measurement ($\pi_o$, *Equation 7* and *Figure 2* in the main text). With the expressions derived in the Appendix 1, Reversible reaction-diffusion model of morphogen dynamics, the thermodynamic cost of generating the steady state morphogen concentration at position $x$ (*Equation 5*) amounts to the product among $\alpha_o$, $\tau(x)$ (*Equation 23*), and $v_{\text{cell}}j_{\text{in}}/l_{\text{cell}}$ (*Equation 33*),

$$
\mathcal{C}(x) = \alpha_o \left(\frac{v_{\text{cell}}j_{\text{in}}}{l_{\text{cell}}k_d}\right)\frac{1}{2}\left(1 + \frac{L}{\lambda}\coth\left[\frac{L}{\lambda}\right] - \frac{L - x}{\lambda}\tanh\left[\frac{L - x}{\lambda}\right]\right),
\tag{37}
$$

and the squared relative positional error, which is identical to *Equation 25*, is

$$
\epsilon^2(x) = \frac{\lambda^3}{l_{\text{cell}}x^2}\left(\frac{v_{\text{cell}}j_{\text{in}}}{l_{\text{cell}}k_d}\right)^{-1}\frac{\sinh\left[\frac{L}{\lambda}\right]\cosh\left[\frac{L - x}{\lambda}\right]}{\left(\sinh\left[\frac{L - x}{\lambda}\right]\right)^2}.
\tag{38}
$$

Then, the trade-off product reads

$$
\begin{aligned}
\pi_o(x; \lambda) &= \mathcal{C}(x)\epsilon^2(x) \\
&= \frac{\alpha_o}{2}\frac{\lambda^3}{l_{\text{cell}}x^2}\left(1 + \frac{L}{\lambda}\coth\left[\frac{L}{\lambda}\right] - \frac{L - x}{\lambda}\tanh\left[\frac{L - x}{\lambda}\right]\right)\frac{\sinh\left[\frac{L}{\lambda}\right]\cosh\left[\frac{L - x}{\lambda}\right]}{\left(\sinh\left[\frac{L - x}{\lambda}\right]\right)^2}.
\end{aligned}
\tag{39}
$$

For fixed $x = x_b$, $\pi_o(x_b; \lambda)$ is numerically minimized with respect to $\lambda$ to give rise to the solid black line in **Figure 2c** in the main text. When $L/\lambda \gg 1$, the trade-off product (**Equation 39**) is approximated to

$$\pi_o(x; \lambda) \approx \frac{\alpha_o}{2} \frac{\lambda}{l_{\text{cell}}} \left[ \left( \frac{\lambda}{x} \right) + \left( \frac{\lambda}{x} \right)^2 \right] e^{x/\lambda}. \tag{40}$$

For a given position $x = x_b$, $\partial_\lambda \pi_o(x_b; \lambda)|_{\lambda = \lambda_{\min}(x_b)} = 0$ determines $\lambda_{\min}(x_b) \approx (\sqrt{13} - 1) x_b / 6 \approx 0.43 x_b$ (gray dotted line in **Figure 2c** of the main text).

## Minimum trade-off product of the space-time-averaged measurement for the irreversible reaction-diffusion process

As in the main text, the trade-off product from the space-time-averaged measurement is given by

$$
\begin{aligned}
\pi_T(x; \lambda) &= \mathcal{C}_T \epsilon_T^2(x) \\
&= \alpha_o j_{\text{in}} T \left( \frac{\partial \log \langle m(x) \rangle}{\partial \log x} \right)^{-2} \frac{\sigma_m^2(x)}{\langle m(x) \rangle^2} \\
&= \alpha_o j_{\text{in}} T \underbrace{\frac{\lambda^2}{x^2 \left[ \frac{2 j_{\text{in}}}{k_{\text{d}}} e^{-x/\lambda} \sinh(a/\lambda) \right]^2}}_{= [x \partial_x \langle m(x) \rangle]^{-2}} \\
&\quad \times \underbrace{\frac{4 j_{\text{in}} e^{-x/\lambda}}{T k_{\text{d}}^2} \sinh(a/\lambda) \left( 1 - \frac{\lambda \cosh(a/\lambda) \sinh(a/\lambda) + a}{2 e^{a/\lambda} \lambda \sinh(a/\lambda)} \right)}_{= \sigma_m^2(x)} \\
&= \alpha_o \left( \frac{\lambda}{x} \right)^2 \frac{e^{x/\lambda}}{\sinh(a/\lambda)} \left( 1 - \frac{\lambda \cosh(a/\lambda) \sinh(a/\lambda) + a}{2 e^{a/\lambda} \lambda \sinh(a/\lambda)} \right) \\
&= \alpha_o \left( \frac{\lambda}{x} \right)^2 e^{x/\lambda} \frac{e^{-a/\lambda} + 3 e^{3a/\lambda} - 4(a \lambda^{-1} + 1) e^{a/\lambda}}{2 (e^{2a/\lambda} - 1)^2},
\end{aligned}
\tag{41}
$$

The $\pi_T(x; \lambda)$ minimizing $\lambda$ at $x = x_b$, namely $\lambda_{\min}(x_b)$, is found at the interval $(x_b - a)/2 \leq \lambda_{\min}(x_b) \leq x_b/2$, since

$$\partial_\lambda \pi_T(x_b; \lambda)|_{\lambda = x_b/2} = \alpha_o \frac{a e^{2 + 2a/x_b} \left( 1 + 3 e^{4a/x_b} \right)}{x_b^2 (e^{4a/x_b} - 1)^3} \left( \sinh \left[ \frac{4a}{x_b} \right] - \frac{4a}{x_b} \right) \geq 0, \tag{42}$$

and

$$\partial_\lambda \pi_T(x_b; \lambda)|_{\lambda = (x_b - a)/2} = \alpha_o \frac{8 a^2 e^{2(x_b + a)/(x_b - a)}}{x_b^2 (x_b - a)(e^{4a/(x_b - a)} - 1)^2} \left( \left[ \frac{2a}{x_b - a} \right]^{-1} - \coth \left[ \frac{2a}{x_b - a} \right] \right) \leq 0. \tag{43}$$

For **Figure 3** in the main text, the exact values of $\pi_{T,\min}(x_b)$ and $\lambda_{\min}(x_b)$ were obtained by numerically minimizing **Equation 41** with respect to $\lambda$.

## Appendix 2

### Distributed morphogen source

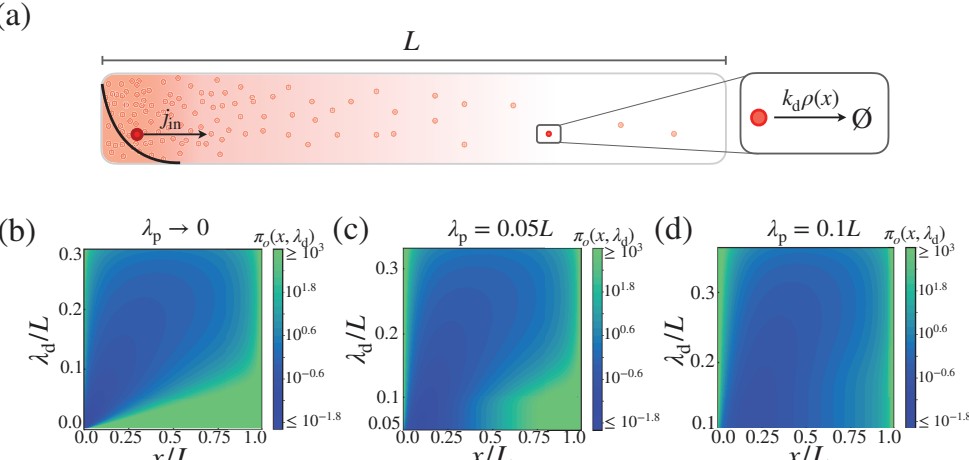

**Appendix 2—figure 1.** Cost-precision trade-off in the morphogen dynamics with distributed synthesis. (**a**) Schematic of the distributed synthesis of the morphogen. The black solid line represents the morphogen synthesis rate which is roughly exponentially distributed with characteristic decay length $\lambda_\mathrm{p}$. (**b–d**) The trade-off product ($\pi_o(x, \lambda_\mathrm{d})$) for (**b**) $\lambda_\mathrm{p} = 0$, (**c**) $\lambda_\mathrm{p} = 0.05L$, and (**d**) $\lambda_\mathrm{p} = 0.1L$. The color scale indicates the trade-off product $\pi_o$ normalized by $\alpha_o L/l_\mathrm{cell}$. The characteristic length scale $\lambda_\mathrm{d}$ is defined as $\lambda_\mathrm{d} \equiv \int_0^L dx(\rho_\mathrm{ss}(x) - \rho_\mathrm{ss}(L))/(\rho_\mathrm{ss}(0) - \rho_\mathrm{ss}(L))$.

We consider a case where the source of morphogen production is distributed over the space in the irreversible reaction-diffusion model of morphogen dynamics. The space-time evolution of the morphogen concentration is described by

$$\partial_t \rho(x,t) = D\partial_x^2 \rho(x,t) - k_\mathrm{d}\rho(x,t) + J_\mathrm{in}(x), \tag{44}$$

with the boundary conditions $\partial_x \rho(x,t)|_{x=0,L} = 0$. The production term, $J_\mathrm{in}(x)$ is assumed to be exponentially distributed with a characteristic length $\lambda_\mathrm{p}$,

$$J_\mathrm{in}(x) = \frac{j_\mathrm{in}}{\lambda_\mathrm{p}} \frac{\cosh\left[\frac{L-x}{\lambda_\mathrm{p}}\right]}{\sinh\left[\frac{L}{\lambda_\mathrm{p}}\right]}, \tag{45}$$

so that $\int_0^L J_\mathrm{in}(x)dx = j_\mathrm{in}$. The solution to **Equation 44** in the Laplace domain is given by

$$\hat{\rho}(x,s) = \frac{j_\mathrm{in}}{s\sqrt{D(k_\mathrm{d}+s)}} \frac{D}{D - (k_\mathrm{d}+s)\lambda_\mathrm{p}^2} \left( \frac{\cosh\left[\frac{L-x}{\sqrt{D/(k_\mathrm{d}+s)}}\right]}{\sinh\left[\frac{L}{\sqrt{D/(k_\mathrm{d}+s)}}\right]} - \frac{\lambda_\mathrm{p}\sqrt{k_\mathrm{d}+s}}{\sqrt{D}} \frac{\cosh\left[\frac{L-x}{\lambda_\mathrm{p}}\right]}{\sinh\left[\frac{L}{\lambda_\mathrm{p}}\right]} \right). \tag{46}$$

By following a procedure similar to that shown in the section Reversible reaction-diffusion model of morphogen dynamics, we can obtain the following expressions for the steady state profile ($\rho_\mathrm{ss}(x)$, **Equation 19**), and the precision ($\epsilon^2(x)$, **Equation 24**):

$$\rho_\mathrm{ss}(x) = \frac{j_\mathrm{in}}{\sqrt{Dk_\mathrm{d}}} \frac{\lambda^2}{\lambda^2 - \lambda_\mathrm{p}^2} \left( \frac{\cosh\left[\frac{L-x}{\lambda}\right]}{\sinh\left[\frac{L}{\lambda}\right]} - \frac{\lambda_\mathrm{p}}{\lambda} \frac{\cosh\left[\frac{L-x}{\lambda_\mathrm{p}}\right]}{\sinh\left[\frac{L}{\lambda_\mathrm{p}}\right]} \right), \tag{47}$$

$$\epsilon^2(x) = \frac{\rho_{ss}(x)}{v_{cell}(x\partial_x\rho_{ss}(x))^2} = \frac{k_d\lambda}{v_{cell}j_{in}} \frac{(\lambda^2 - \lambda_p^2)\left(\frac{\cosh\left[\frac{L-x}{\lambda}\right]}{\sinh\left[\frac{L}{\lambda}\right]} - \frac{\lambda_p}{\lambda}\frac{\cosh\left[\frac{L-x}{\lambda_p}\right]}{\sinh\left[\frac{L}{\lambda_p}\right]}\right)}{x^2\left[\left(\frac{\left[\frac{L-x}{\lambda}\right]}{\sinh\left[\frac{L}{\lambda}\right]} - \frac{\sinh\left[\frac{L-x}{\lambda_p}\right]}{\sinh\left[\frac{L}{\lambda_p}\right]}\right)\right]^2},$$ (48)

with $\lambda \equiv \sqrt{D/k_d}$. Next, the characteristic timescale to approach the steady state profile ($\tau(x)$, **Equation 22**) can be written as

$$\tau(x) = \mathcal{N}/\mathcal{M},$$ (49)

with

$$\mathcal{N} \equiv \left(1 - 3\frac{\lambda_p^2}{\lambda^2} + \frac{L}{\lambda}\left(1 - \frac{\lambda_p^2}{\lambda^2}\right)\coth\left[\frac{L}{\lambda}\right]\right)\frac{\cosh\left[\frac{L-x}{\lambda}\right]}{\sinh\left[\frac{L}{\lambda}\right]} + 2\frac{\lambda_p^3}{\lambda^3}\frac{\cosh\left[\frac{L-x}{\lambda_p}\right]}{\sinh\left[\frac{L}{\lambda_p}\right]} - \frac{L-x}{\lambda}\left(1 - \frac{\lambda_p^2}{\lambda^2}\right)\frac{\sinh\left[\frac{L-x}{\lambda}\right]}{\sinh\left[\frac{L}{\lambda}\right]},$$

and

$$\mathcal{M} \equiv 2k_d\left(-1 + \frac{\lambda_p^2}{\lambda^2}\right)\left(\frac{\lambda_p}{\lambda}\cosh\left[\frac{x}{\lambda_p}\right]\coth\left[\frac{L}{\lambda_p}\right] - \frac{\lambda_p}{\lambda}\sinh\left[\frac{x}{\lambda_p}\right] - \frac{\cosh\left[\frac{L-x}{\lambda}\right]}{\sinh\left[\frac{L}{\lambda}\right]}\right).$$

Then, the trade-off product is expressed as

$$\pi(x;\lambda,\lambda_p) = \alpha_o\left(\frac{v_{cell}j_{in}}{l_{cell}}\right)\tau(x)\epsilon^2(x),$$ (50)

with the proportionality constant $\alpha_o$. For fixed $x = x_b$, $L$, and $\lambda_p$, the trade-off product can be minimized by tuning $\lambda$ to the optimal value, $\lambda_{min}(x_b)$. As $\lambda_p$ increases, the optimal trade-off product and the associated optimal characteristic length $\lambda_{min}(x)$ decrease (**Appendix 2—figure 1**). In **Appendix 2—figure 1**, the characteristic length scale $\lambda_d$ is defined as $\lambda_d \equiv \int_0^L dx(\rho_{ss}(x) - \rho_{ss}(L))/(\rho_{ss}(0) - \rho_{ss}(L))$.

## Dynamics on a sphere

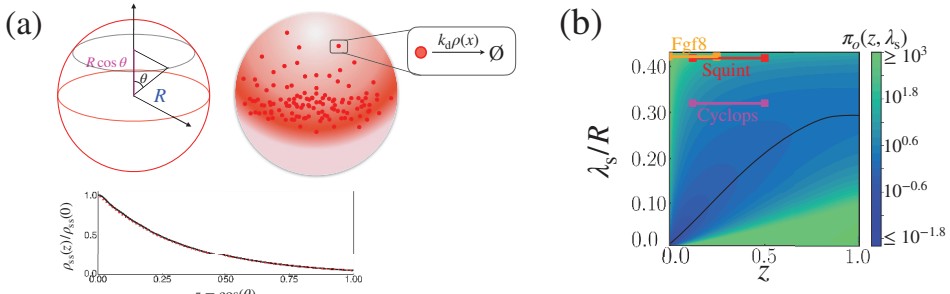

**Appendix 2—figure 2.** Cost-precision trade-off of the morphogen dynamics on a sphere. (a) (Top) Schematic of the morphogen synthesized from the equator of the spherical shell (red region). (Bottom) The relative steady state concentration profile $\rho_{ss}(z)$ when $R^2/\lambda^2 = 10$. The solid black line was computed from evaluating $\rho_{ss}(z) = \lim_{s\to 0} s\hat{\rho}(z,s)$ up to $l = 84$ (**Equation 53**). The dotted red line was obtained by numerically solving **Equation 51**. (b) The trade-off product ($\pi_o(z,\lambda_s)$) for pairs of target position $z$ and characteristic length scale $\lambda_s$ values, normalized by $\alpha_o(2\pi R/l_{cell})$. The characteristic length scale $\lambda_s$ is defined as $\lambda_s \equiv \int_0^1 dz(\rho_{ss}(z) - \rho_{ss}(1))/(\rho_{ss}(0) - \rho_{ss}(1))$. The parameters for the naturally occurring morphogen profiles are further described in Appendix 3, Length scales of Cyclops, Squint, and Fgf8, and **Appendix 3—table 1**.

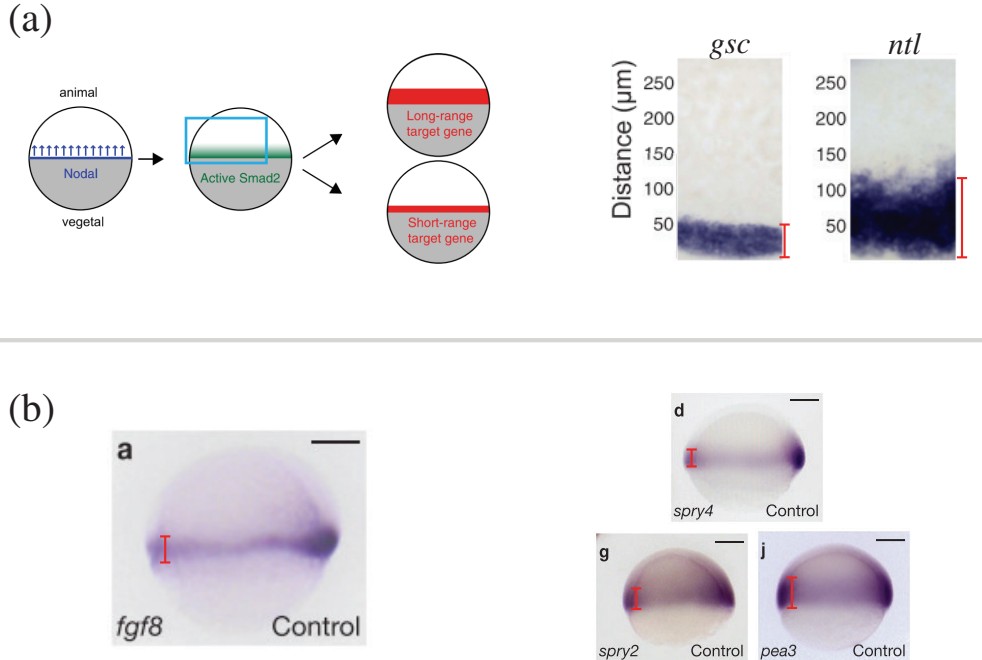

**Appendix 2—figure 3.** Length scales of naturally occurring morphogen profiles in the zebrafish embryo. (**a**) (Left) Schematic of the Nodal ligands Cyclops and Squint patterning the vegetal-animal axis of the zebrafish embryo. The Nodal signals activate Smad2, which affects the expression of short-range and long-range target genes. (Right) The expression profile of Nodal target genes *gsc* and *ntl* obtained by in situ hybridization. The profiles are from the zebrafish embryos 5 hr post fertilization, when the embryo covers roughly 50% of the yolk. The red lines denote the width of the target gene expression domain away from the vegetal margin. (**b**) (Left) Expression of *fgf8* in 60% epiboly whole-mount embryos obtained by in situ hybridization. Animal side of the embryo is to the top, and dorsal is to the right. (Right) The expression profiles of the Fgf8 target genes *spry4*, *spry2*, and *pea3*. The red lines denote the width of the target gene expression domain away from the vegetal margin. Scale bars = 100 $\mu$m. (**A**) is reproduced from Figure 2 *Dubrulle et al., 2015*. (**B**) is reproduced from Figure 1 *Nowak et al., 2011*. Reprinted by permission from Springer Nature: Springer Nature, Nature Cell Biology, Interpretation of the FGF8 morphogen gradient is regulated by endocytic trafficking, Nowak, M., MacHate, A., Yu, S. R., Gupta, M., and Brand, M. 2011, Nat. Cell Biol., 13(2):153–158 Copyright 2011. It is not covered by the CC-BY 4.0 licence and further reproduction of this panel would need permission from the copyright holder.

We consider the morphogen profile formation on the surface of a sphere. The profile formations of Cyclops, Squint, and Fgf8 in the developing zebrafish embryo at ~50% epiboly pertain to this situation. When the concentration of the morphogen in a cell with spherical coordinates $(r, \theta, \phi)$ is denoted by $\rho(r, \theta, \phi)$, with the symmetry along the azimuthal angle and the condition of the morphogen dynamics being confined to the spherical shell of radius $r = R$ (i.e. $\partial_\phi \rho = 0$, $\partial_r \rho = 0$, and $r = R$) the morphogen concentration satisfies the following evolution equation:

$$\frac{\partial \rho(\theta,t)}{dt} = D\left[\frac{1}{R^2 \sin\theta}\frac{\partial}{\partial\theta}\left(\sin\theta\frac{\partial\rho(\theta,t)}{\partial\theta}\right)\right] - k_{\mathrm{d}}\rho(\theta,t) + j_{\mathrm{in}}\delta(\cos\theta), \tag{51}$$

with boundary conditions at $\partial_\theta\rho|_{\theta=0,\pi} = 0$. The morphogen dynamics is fully specified by the system size $R[l_{\mathrm{cell}}]$, the depletion rate $k_{\mathrm{d}}[1/\mathrm{time}]$, the diffusion coefficient $D[l_{\mathrm{cell}}^2/\mathrm{time}]$, and the injection rate $j_{\mathrm{in}}[\mathrm{conc}/\mathrm{time}]$. The last term represents the influx of morphogens due to their synthesis by the 'band' of cells at the equator (*Appendix 2—figure 2a*). With the substitution $z = \cos\theta$ and the initial condition $\rho(z,t) = 0$, the solution of *Equation 51* in the Laplace domain with $\hat{\rho}(z,s) = \int_0^\infty dt e^{-st}\rho(z,t)$, can be obtained by solving

$$\frac{\partial}{\partial z}\left((1-z^2)\frac{\partial\hat{\rho}(z,s)}{\partial z}\right) - \frac{R^2(k_{\mathrm{d}}+s)}{D}\hat{\rho}(z,s) + \frac{R^2 j_{\mathrm{in}}}{Ds}\delta(z) = 0. \tag{52}$$

The solution to the differential equation of this type can be expressed as $\hat{\rho}(z,s) = \sum_{l=0}^\infty c_l(s)P_l(z)$, where $P_l(z)$ $(l = 0,1,\ldots)$ is the Legendre polynomial, satisfying $\mathcal{L}P_l(x) \equiv \frac{d}{dx}\left[(1-x^2)\frac{dP_l(x)}{dx}\right] = -l(l+1)P_l(x)$, with the orthogonality condition $\int_{-1}^1 P_l(x)P_m(x)dx = \frac{2}{2l+1}\delta_{lm}$. To find the coefficient $c_l(s)$, we rearrange *Equation 52* into

$$-\frac{R^2 j_{\mathrm{in}}}{Ds}\delta(z) = -\sum_{l=0}^\infty\left(l(l+1) + \frac{R^2(k_{\mathrm{d}}+s)}{D}\right)c_l(s)P_l(z).$$

By using the orthogonality condition of Legendre polynomials, we obtain the solution of *Equation 52*,

$$\hat{\rho}(z,s) = \frac{R^2 j_{\mathrm{in}}}{Ds}\sum_{l=0}^\infty\frac{(2l+1)}{2\left(\frac{R^2(k_{\mathrm{d}}+s)}{D} + l(l+1)\right)}P_l(0)P_l(z). \tag{53}$$

The steady state concentration can be expressed as $\rho_{\mathrm{ss}}(z) = \lim_{s\to 0}s\hat{\rho}(z,s)$.

The average relaxation time can be calculated using $\tau(z) \equiv \langle t\rangle = \int_0^\infty t\left(-\frac{dR(t)}{dt}\right)dt = \int_0^\infty R(t)dt = \lim_{s\to 0}\hat{R}(z,s)$, which yields

$$\tau(z) = k_{\mathrm{d}}^{-1}\mathcal{T}(z,\lambda/R), \tag{54}$$

with $\lambda \equiv \sqrt{D/k_{\mathrm{d}}}$ and

$$\mathcal{T}(z,\lambda/R) = \frac{R^2}{\lambda^2}\cdot\frac{\sum_{l=0}^\infty\frac{(2l+1)}{2\left(R^2/\lambda^2 + l(l+1)\right)^2}P_l(0)P_l(z)}{\sum_{l=0}^\infty\frac{(2l+1)}{2\left(R^2/\lambda^2 + l(l+1)\right)}P_l(0)P_l(z)}.$$

The number of morphogen molecules produced from the cells at the equator per unit time is $v_{\mathrm{cell}}(2\pi R/l_{\mathrm{cell}})j_{\mathrm{in}}$. Thus, the thermodynamic cost to approach the steady state concentration at $z$ is

$$C(z) = \alpha_o 2\pi R\frac{v_{\mathrm{cell}}j_{\mathrm{in}}}{l_{\mathrm{cell}}k_{\mathrm{d}}}\mathcal{T}(z,\lambda/R), \tag{55}$$

where $\alpha_o$ is the proportionality constant.

Next, using $\rho_{\mathrm{ss}}(z) = \lim_{s\to 0}s\hat{\rho}(z,s)$ (*Equation 53*), one obtains the local precision of the morphogen profile at position $z$,

$$\epsilon^2(z) = \frac{\rho_{\mathrm{ss}}(z)}{v_{\mathrm{cell}}(z\partial_z\rho(z))^2} = \frac{k_{\mathrm{d}}}{v_{\mathrm{cell}}j_{\mathrm{in}}}\mathcal{E}(z,\lambda/R), \tag{56}$$

where

$$\mathcal{E}(z,\lambda/R) \equiv \frac{\frac{\lambda^2}{R^2}\sum_{l=0}^\infty\frac{(2l+1)}{2\left(R^2/\lambda^2 + l(l+1)\right)}P_l(0)P_l(z)}{z^2\left(\sum_{l=0}^\infty\frac{(2l+1)}{2\left(R^2/\lambda^2 + l(l+1)\right)}P_l(0)\partial_z P_l(z)\right)^2}.$$

Finally, the trade-off product, which is plotted in **Appendix 2—figure 2b**. Eb as a function of $z$ and $\lambda_s$, is defined as

$$\pi(z;\lambda/R) \equiv C(z) \times \epsilon^2(z) = \alpha_o \frac{2\pi R}{l_{\text{cell}}} \mathcal{T}(z,\lambda/R)\mathcal{E}(z,\lambda/R). \tag{57}$$

The length scale $\lambda_{\text{s}}$ used in the plot is defined as $\lambda_{\text{s}} \equiv \int_{z=0}^{z=1} dz(\rho_{\text{ss}}(z) - \rho_{\text{ss}}(1))/(\rho_{\text{ss}}(0) - \rho_{\text{ss}}(1))$. At the target position $z = z_{\text{b}}$, the trade-off product can be minimized by tuning $\lambda$ to the optimal value $\lambda_{\text{min}}$, which is plotted in black in **Appendix 2—figure 2b**.

## Appendix 3

### Length scales of Bcd, Wg, Hh, and Dpp

Here we explain the length scales associated with the naturally occurring morphogen profiles of *Drosophila* (*Figure 2—figure supplement 1*, *Appendix 3—table 1*).

- Bcd. The characteristic decay length of Bcd was set to $\lambda_{\mathrm{Bcd}} = 100\,\mu\mathrm{m}$ (*Houchmandzadeh et al., 2002*). The length of the embryo across the anterior-posterior axis is 475 μm, and the Bcd profile decays exponentially starting at 50 μm from the anterior end (*Houchmandzadeh et al., 2002*). Thus, we set the system length to $L = 425\,\mu\mathrm{m}$. The sensor radius $a = 2.6\,\mu\mathrm{m}$ was obtained from the image of nuclei in Figure 3A of *Gregor et al., 2007*. The Bcd target Hb expression displays a sharp decline in the range of (190–238) μm (Figure 3C of *Perry et al., 2012*). Since the Bcd profile decays exponentially starting at 50 μm from the anterior end, we set $x_{\mathrm{b}} = (140 - 188)\,\mu\mathrm{m}$.
- Wg. The characteristic decay length of Wg was set to $\lambda_{\mathrm{Wg}} = 6\,\mu\mathrm{m}$ (*Kicheva et al., 2007*). The overall size of the DV axis patterned by Wg was set to $L = 70\,\mu\mathrm{m}$, based on the Hoechst staining image in Figure 1E of *Chaudhary et al., 2019*. Although the Wg dynamics reported in *Kicheva et al., 2007* is not from the endogenous concentration profile, we assume that the wild type profile forms with similar length and time scales. While Wg induces the expression of multiple genes, including *distalless* and *vestigial*, we picked *sens* for our analysis because its expression profile was quantitatively shown to undergo a sharp transition. *sens* expression decreases rapidly at (15–25) μm away from the DV boundary (Figure 1G of *Bakker et al., 2020*). Assuming that Wg is secreted by the two stripes of cells with radius 2 μm at the DV boundary (Figure 2A from *Chaudhary et al., 2019*), we set the target boundary to $x_{\mathrm{b}} = (11 - 21)\,\mu\mathrm{m}$.
- Hh. The characteristic decay length of Hh was set to $\lambda_{\mathrm{Hh}} = 8\,\mu\mathrm{m}$ (Figure S2B of *Wartlick et al., 2011*). The overall size of the domain patterned by Hh was set to $L = 100\,\mu\mathrm{m}$ (Figure 6C of *Torroja et al., 2004*). The Hh target genes include *en*, *col*, and *dpp*, whose expression profiles were obtained from Figure 5 of *Torroja et al., 2004*. The expression domains of *en*, *col*, and *dpp*, respectively, span 15 μm, 25 μm, and 30 μm away from the Hh producing cells. Accordingly, the range of boundary positions associated with Hh was set to $x_{\mathrm{b}} = (15 - 30)\,\mu\mathrm{m}$.
- Dpp. The characteristic decay length of Dpp was set to $\lambda_{\mathrm{Dpp}} = 20\,\mu\mathrm{m}$ (*Kicheva et al., 2007*). The overall size of the domain patterned by Dpp was set to $L = 80\,\mu\mathrm{m}$ (Figure 2H of *Bakker et al., 2020*). The boundary of Dpp target *salm* expression was set to $x_{\mathrm{b}} = (36 - 54)\,\mu\mathrm{m}$ (Figure 2H of *Bakker et al., 2020*). The radius of the sensor for Dpp and Hh mediated patterning was set to $a = 2\,\mu\mathrm{m}$, same as for Wg, since they diffuse through the same wing imaginal disk. While we only focus on the boundary defined by *salm* for simplicity, other genes affected by Dpp, including *omb*, *brk*, and *dad*, also display changes in their spatial expression profiles at locations similar to that of *salm* (*Bakker et al., 2020*).

### Length scales of Cyclops, Squint, and Fgf8

In zebrafish embryos, the morphogens Cyclops, Squint, and Fgf8 are responsible for the induction of the endoderm and the mesoderm. All three morphogens are synthesized at the embryonic margin, and diffuse toward the animal pole (*Appendix 2—figure 3a*). For simplicity, we will only consider zebrafish embryos around the ~50% epiboly stage, when the embryo encompasses the top half of the spherical yolk. The length scales associated with Cyclops, Squint, and Ffg8 are also given in *Appendix 3—table 1*.

- Cyclops and Squint. The diffusion coefficient ($D$) and depletion rate ($k_{\mathrm{d}}$) of Cyclops and Squint have been measured through the ectopic expressions of the fluorescently tagged constructs. The resulting characteristic decay lengths ($\lambda \equiv \sqrt{D/k_{\mathrm{d}}}$) are $\lambda_{\mathrm{cyclops}} = 76\,\mu\mathrm{m}$ and $\lambda_{\mathrm{squint}} = 178\,\mu\mathrm{m}$ (*Müller et al., 2012*). The target boundary of Nodal signaling was set by the expression domain width of the target genes *gsc* (50 μm) and *ntl* (125 μm) (Figure 1f of *Dubrulle et al., 2015*). The width of the domain of Nodal morphogen production was assumed to be 25 μm, following the kinetic model describing Nodal dynamics in *Dubrulle et al., 2015*. The radius of the zebrafish embryo was set to $R = 200\,\mu\mathrm{m}$ (Figure 1 of *Nowak et al., 2011*). Overall, for Nodal morphogens Cyclops and Squint, $z_{\mathrm{b}}$ ranges in [(50 μm – 25 μm)/200 μm, (125 μm – 25 μm)/200 μm] = [0.125, 0.5].

- Fgf8. The characteristic decay length of Fgf8, $\lambda_{\text{fgf8}} = 197\,\mu\text{m}$, was measured from the ectopic expression of the fluorescently tagged constructs (*Yu et al., 2009*). Fgf8 affects the expression of target genes such as *spry4*, *spry2*, and *pea3*. The target boundary of Fgf8 was set by the expression domain width of the target genes *spry4* (70 μm), *spry2* (85 μm), and *pea3* (120 μm) (Figure 1 of *Nowak et al., 2011*). Subtracting the width of the Fgf8 expression domain (70 μm) and dividing by the radius of the embryo (200 μm) amount to $z_{\text{b}}$ in the range of $[(70\mu\text{m} - 70\mu\text{m})/200\mu\text{m}, (120\mu\text{m} - 70\mu\text{m})/200\mu\text{m}] = [0.0, 0.25]$.

**Appendix 3—table 1.** Various length scales associated with the naturally occurring morphogen profiles found in the fruit fly (*) and zebrafish (†) embryos.

For each plot of the morphogen profile in *Figure 2—figure supplement 1*, the solid red line denotes the exponentially decaying profile with the associated characteristic length $\lambda$. Further details on how each value was obtained from the associated reference are provided in the text of Appendix 3.

| | $L^*$ or $R^\dagger$ (μm) | $\lambda[\equiv \sqrt{D/k_{\text{d}}}]$ (μm) | Range of $x_{\text{b}}^*$ or $z_{\text{b}}^\dagger$ (μm) | $a$ (μm) |
|---|---|---|---|---|
| Bcd* | 425 *Houchmandzadeh et al., 2002* | 100 *Houchmandzadeh et al., 2002* | 140–188 *Perry et al., 2012* | 2.6 *Gregor et al., 2007* |
| Wg* | 70 *Chaudhary et al., 2019* | 6 *Kicheva et al., 2007* | 11–21 *Bakker et al., 2020* | 2 *Chaudhary et al., 2019* |
| Hh* | 100 *Torroja et al., 2004* | 8 *Wartlick et al., 2011* | 15–30 *Torroja et al., 2004* | 2 *Chaudhary et al., 2019* |
| Dpp* | 80 *Bakker et al., 2020* | 20 *Kicheva et al., 2007* | 36–54 *Bakker et al., 2020* | 2 *Chaudhary et al., 2019* |
| Cyclops† | 200 *Nowak et al., 2011* | 76 *Dubrulle et al., 2015* | 25–100 *Dubrulle et al., 2015* | 10 *Yu et al., 2009* |
| Squint† | 200 *Nowak et al., 2011* | 178 *Dubrulle et al., 2015* | 25–100 *Dubrulle et al., 2015* | 10 *Yu et al., 2009* |
| Fgf8† | 200 *Nowak et al., 2011* | 197 *Yu et al., 2009* | 0–50 *Nowak et al., 2011* | 10 *Yu et al., 2009* |

## Time scales of Bcd, Wg, Hh, and Dpp

The expression for the variance of the space-time-averaged signal, $\sigma_m^2(x)$ (*Equation 11*), is derived with the assumption of a long measurement time, $T \gg k_{\text{d}}^{-1}$ (Appendix 1 of *Fancher and Mugler, 2020*). Additionally, in order for $Tj_{\text{in}}$ to represent the total cost of establishing and maintaining the morphogen profile, we require the measurement time to be much longer than the time scale of reaching the steady state (i.e. $T \gg \tau(x_{\text{b}})$). For Bcd, Wg, Hh, and Dpp, $L \gg \lambda$ and $x_{\text{b}}/\lambda \approx 2$, leading to the approximate expression for the characteristic time of $\tau(x_{\text{b}}) \approx 1.5k_{\text{d}}^{-1}$ (*Equation 4*). We assume that the maximum measurement time, $T_{\text{max}}$, is set by the cell doubling time, after which the fates of the daughter cells must be re-established.

- Bcd: $T_{\text{max}}[\approx 10\,\text{min}(\textit{Foe and Alberts, 1983})] < k_{\text{d}}^{-1}[\approx 1\,\text{hr}(\textit{Drocco et al., 2011})]$.
- Wg: $T_{\text{max}}[\approx 5\,\text{hr}(\textit{Martín et al., 2009})] \gg k_{\text{d}}^{-1}[\approx 15\,\text{min}(\textit{Kicheva et al., 2007})]$.
- Hh: $T_{\text{max}}[\approx 5\,\text{hr}(\textit{Martín et al., 2009})] \gg k_{\text{d}}^{-1}[\approx 5\,\text{min}(\textit{Nahmad and Stathopoulos, 2009})]$.
- Dpp: $T_{\text{max}}[\approx 5\,\text{hr}(\textit{Martín et al., 2009})] > k_{\text{d}}^{-1}[\approx 1.4\,\text{hr}(\textit{Kicheva et al., 2007})]$.

For Dpp, Hh, and Wg, the space-time-averaged measurements may be performed for sufficiently long durations to represent the cost-precision trade-off by $\pi_T$ (*Equation 14*). For Bcd, $\pi_o$, which is formulated for the point measurement, may better represent the pertinent cost-precision trade-off.

