## [Decision Letter]

**Acceptance summary:**

This manuscript presents a theoretical investigation on cost-precision relations in the formation of biological signaling profiles (also known as morphogen gradients) that are responsible for the proper development of biological organisms. It is argued that the thermodynamic cost is an important factor in the morphogen gradient formation, and several signaling molecule gradients in *Drosophila* appear to be optimized according to the presented theoretical estimates. This is one of the first quantitative studies of the optimization of the development morphogen gradients, and its main arguments are strong and convincing. One of the main conclusions of this work is that the complex task of biological signaling processes seems to be optimized, and the paper should be interesting to a wide range of scientists working on both experimental and theoretical aspects of understanding the fundamental origin of optimization in biological processes.

**Decision letter after peer review:**

Thank you for submitting your article "Cost-precision trade-off relation determines the optimal morphogen gradient for accurate biological pattern formation" for consideration by *eLife*. Your article has been reviewed by 3 peer reviewers, and the evaluation has been overseen by a Reviewing Editor and Naama Barkai as the Senior Editor. The following individuals involved in review of your submission have agreed to reveal their identity: Mariela Petkova (Reviewer #2); Anatoly Kolomeisky (Reviewer #3).

Essential Revisions:

1) The main assumption in the paper is that the cost of creating a morphogen gradient is something that the biological system cares about. However, it is not shown that the cost of creating the morphogen gradient is a significant portion of the total energy budget of the biological system, and therefore potentially necessary to be optimized. The choice of trade-off function was presented without supporting arguments that either reflect biological expectations or demonstrate that the results are robust to other trade-off constructs. Importantly, the paper argues that the agreement between the optimal gradient shape predicted by the trade-off framework and the measurements in real biological systems indicates that thermodynamic cost is a key constraint determining the morphogen shape. However, perturbed systems such as increasing and decreasing the dosage of Bicoid still produce viable embryos (Liu et al., PNAS, 2013). The latter observation suggests that both producing different amount of morphogen molecules and employing regulatory mechanisms to correct the effects of perturbations are not energetically cost-limiting to an embryo. Justifying that this cost is significant with respect to the cells' energy budget would greatly aid the manuscript.

2) Similarly, the authors choose a product of thermodynamic cost and the squared relative error in the position as a trade-off function that they analyze. My question is how robust are the obtained results? The choice of the trade-off function is not unique and it does not have any fundamental foundation, to the best of my understanding. For instance, one defines X as a thermodynamic cost and Y as relative error in position, then not only X*Y, but also (X+Y), X*Y*Y, X*X*Y and many other combinations also might be viewed as trade-off functions. But they would lead to a different prediction on the optimized minimal length and other related issues. Then how to choose and why?

3) The work constraints the patterning problem to a gradient which produces a single target boundary, but validates the results with measurements in systems where gradients are known to regulate many downstream patterns. The result of the trade-off analysis is that the location of the target boundary determines the shape of the gradient. How is this reconciled with the understanding that, for example, the Bicoid gradient is read out by multiple downstream genes along the entire length of the fly embryo? (Hannon et al., *eLife*, 2017).

4) The description of the error above Equation 1 is somewhat confusing in two different ways: i) definitions are given in terms of mathematical variables like N and the subscript i that are not explicitly explained. The reader can figure out the intended meaning via later discussions, but it would be helpful to make these terms clear from the start. ii) More broadly, the notion of "measurement" used in the paper is a little abstract, and is likely to confuse biologically-oriented readers. Before delving into the mathematical definition of measurement errors, could the authors provide a summary of how measurements could hypothetically be carried out via concrete biological mechanisms? For example, they mention that space-time-averaged measurements might be carried out by cell surface receptors. But what mechanisms could lead to instantaneous point measurements?

5) In line 181 the authors state that "the two limiting models can be effectively minimized by similar λ values at a given position is of great significance." However comparing Figures 2(c) and 3(c) it is a hard to directly see this, because in one case the y-axis is normalized as λ/L and in the other case as λ/a. Could the authors somehow make this similarity in predicted λ values clearer? Maybe adding a twin y axis scale to one of the figures, showing λ in terms of the other normalization?

6) The discussion of the reversible model in the appendix is quite nice, particularly how you can derive all the irreversible results directly from the reversible model by taking an appropriate limit. Particularly interesting is that the cost defined in terms of entropy production rate times characteristic time, if normalized by ln(γ) [using Equation 33] would seem to give you the morphogen production cost of Equation 37, up to a factor of α_0_, assuming the irreversible limit where ln(γ) -> infinity. So it seems that the entropy-based cost is closely related to the other definition of cost, and the authors could consider highlighting this fact in the main text. Is there any qualitative way of understanding why ln(γ) is the appropriate normalization factor?

7) All derivations are mode for the finite length L. However, as argued in the specific calculations L>>\λ, so why not to derive the results for L>>1. In this case, all the formulas are much simpler and there is a clear physical interpretation for all the quantities. Why the bulkier and less transparent expressions are used in the text? If authors wanted to explain things better to a broader readership and to emphasize the physics they better use a simple analytical framework. Or they need to explain why they used a finite L.

8) It would be better to describe fully how the characteristic lengths were estimated. This is because the reported numbers do not fully agree with what is reported in experiments. For example, for BCD Berezhkovskii and Shvartsman report λ=75 nm, not 100nm as given in this paper. Why are such different results are reported?

*Reviewer #1:*

There has been a recent surge of interest in trying to understand the tradeoff between the precision achieved by non-equilibrium processes and the underlying thermodynamic costs. One of the most interesting arenas for this exploration has been biology, and in the current paper, the authors have discovered a novel form of this tradeoff in the context of pattern formation during animal development.

Their main focus is a 1D mathematical model describing how morphogens are produced, diffuse, and depleted within the cell, forming a gradient that divides the cell into two spatial regions with different subsequent developmental trajectories. Establishing a sharp boundary between these regions comes at a certain cost in terms of morphogen production, and the authors provide an elegant theoretical argument that the boundary precision and cost bounded by an "uncertainty principle" type relationship, where their product is always greater than a certain system-specific quantity. The result is reminiscent of, but qualitatively different from, the so-called thermodynamic uncertainty principle that has generated a lot of excitement in the non-equilibrium statistical physics community in the last few years.

Finding such a tradeoff is interesting in itself, but is it biologically relevant? In this respect the paper makes two striking claims: (i) making the system optimally cost effective in achieving a certain precision predicts a certain characteristic decay length for the morphogen concentration profile; (ii) this prediction is directly testable using data from earlier *Drosophila* development experiments. The authors' analysis shows how four morphogens (Hh, Wg, Bcd, Dpp) to a good approximation fulfill the criteria for optimality in this sense. This conclusion seems robust to different mechanisms by which cells can "measure" the morphogen profile to translate it into cell fate decisions, whether these measurements are done instantaneously at precise locations or averaged over time and space.

Overall the results of the paper are well supported by the theoretical analysis and data. The optimality of the four *Drosophila* morphogens suggests that cost-effectiveness has played an evolutionary role in shaping morphogenesis, at least in this organism. One complicating factor in this conclusion (which to their credit, is explored by the authors) is that optimality may not be achieved in systems with more complex geometries, like zebrafish. How and when thermodynamic costs are decisive in biological contexts is a fascinating question for future study. This paper is a significant contribution to that discussion and a wonderful example of physics-inspired theory helping us interpret well-studied biological systems in a completely new way.

*Reviewer #2:*

This paper assumes that the thermodynamic cost of producing patterning molecules constraints the events which occur during development. It takes a minimalistic view of a general class of problems in developmental systems: how a morphogen gradient specifies a boundary location in a downstream gene expression pattern. The authors evaluate the cost of generating a morphogen gradient against the precision with which the gradient can specify the target boundary. Making more morphogen molecules costs more, but reduces variability in the gradient and improves the precision with which a target boundary can be set up. The authors develop a trade-off framework for two gradient readout limits: instantaneous and spatiotemporally averaged. In both cases, the optimal solution for the shape of the morphogen gradient is consistent with experimental measurements in multiple biological systems. The key claim of the work is that this agreement between prediction and measurement is evidence that developing systems employ such trade-offs.

Strengths:

The trade-off framework developed in this paper has three major strengths. First, it is simple and generalizable to multiple systems and geometries: developing embryos in multiple species as well as gradient-mediated tissue patterning. Second, it depends only on geometric features such as length, boundary location and gradient shape. These can be readily measured in real biological systems and thus test the framework's predictions. Lastly, it captures how the dynamics of reading out the morphogen gradient affects the trade-off between morphogen cost and target boundary precision. The key result of the predictions is that the same shape of the morphogenic gradient optimizes two readout strategies which are at the two extremes of biological function – instantaneous readout, and spatiotemporal averaging.

Weaknesses:

The weakness in this paper lies in reconciling the work's assumptions and results with their interpretations in the context of complex biological systems.

The main assumption in the paper is that the cost of creating a morphogen gradient is something that the biological system cares about. However, it is not shown that the cost of creating the morphogen gradient is a significant portion of the total energy budget of the biological system, and therefore potentially necessary to be optimized.

Next, the work constraints the patterning problem to a gradient which produces a single target boundary, but validates the results with measurements in systems where gradients are known to regulate many downstream patterns. The result of the trade-off analysis is that the location of the target boundary determines the shape of the gradient. How is this reconciled with the understanding that, for example, the Bicoid gradient is read out by multiple downstream genes along the entire length of the fly embryo? (Hannon et al.,2017)

Finally, the authors present only one possibility for the trade-off function and evaluate it against measurements in unperturbed systems. The choice of trade-off function was presented without supporting arguments that either reflect biological expectations or demonstrate that the results are robust to other trade-off constructs. Importantly, the paper argues that the agreement between the optimal gradient shape predicted by the trade-off framework and the measurements in real biological systems indicates that thermodynamic cost is a key constraint determining the morphogen shape. However, perturbed systems such as increasing and decreasing the dosage of Bicoid still produce viable embryos (Liu et al., 2013). The latter observation suggests that both producing different amount of morphogen molecules and employing regulatory mechanisms to correct the effects of perturbations are not energetically cost-limiting to an embryo.

References:

Hannon, C.E., Blythe, S.A. and Wieschaus, E.F., 2017. Concentration dependent chromatin states induced by the bicoid morphogen gradient. e*Life*, 6, p.e28275.

Liu, F., Morrison, A.H. and Gregor, T., 2013. Dynamic interpretation of maternal inputs by the *Drosophila* segmentation gene network. Proceedings of the National Academy of Sciences, 110(17), pp.6724-6729

*Reviewer #3:*

This is a very elegant theoretical investigation on the microscopic features of the formation of biological signaling profiles. A comprehensive theoretical model that estimates the thermodynamic cost and the accuracy is presented. The theoretical predictions are utilized for analyzing several morphogens in *Drosophila*. The strengths of the paper are the following: (i) it addresses a fundamental question of optimization of biological systems from the fundamental point of view, (ii) the approach is fully quantitative, (iii) the theoretical framework is explored for estimating the optimization of morphogen gradients in the real biological system, and (iv) several extensions for more realistic situations of the morphogen gradients formation are outlined. All these features make me believe that the conclusions of the paper are justified. However, despite the fact that the paper is well written and the overall results seem convincing, there are several issues that need to be clarified. The weaknesses (not very significant) include: (i) the estimates of lengths do not fully agree with some of the reported lengths in the literature, (ii) the robustness of the cost-precision trade-off relation is not critically evaluated, and (iii) some references to the existing literature are missing.

---

## [Author Response]

Essential Revisions:1) The main assumption in the paper is that the cost of creating a morphogen gradient is something that the biological system cares about. However, it is not shown that the cost of creating the morphogen gradient is a significant portion of the total energy budget of the biological system, and therefore potentially necessary to be optimized. The choice of trade-off function was presented without supporting arguments that either reflect biological expectations or demonstrate that the results are robust to other trade-off constructs. Importantly, the paper argues that the agreement between the optimal gradient shape predicted by the trade-off framework and the measurements in real biological systems indicates that thermodynamic cost is a key constraint determining the morphogen shape. However, perturbed systems such as increasing and decreasing the dosage of Bicoid still produce viable embryos (Liu et al., PNAS, 2013). The latter observation suggests that both producing different amount of morphogen molecules and employing regulatory mechanisms to correct the effects of perturbations are not energetically cost-limiting to an embryo. Justifying that this cost is significant with respect to the cells' energy budget would greatly aid the manuscript.

Thank you for raising these important points. As demonstrated in Liu et al., it is difficult to predict the effect of perturbing the Bcd profile on the organism’s fitness, because the embryonic anterior patterning is governed by a large network of co-regulating signaling molecules. Additionally, as detailed in the revised manuscript, the thermodynamic cost of generating the Bcd morphogen profile is only a small fraction of the total energy budget of embryogenesis. However, the energy budget of developing systems remains largely uncharacterized, and there is certainly a lack of understanding on how much of the cost can be attributed to “housekeeping processes”, as opposed to the cost of generating new spatial structures (Rodenfels et al., *Dev. Cell* 2019; Rodenfels el al., *MBoC* 2020; Song et al., *Curr. Biol.* 2019). The generation of each morphogen profile is an indispensable developmental process that contributes to the total energy budget, if only by a small amount. In this light, our theory proposes a quantitative framework to evaluate the cost-precision trade-off of individual morphogen profiles, which allows us to show that the morphogen profiles of fruit fly development are nearly optimal, as opposed to those of the zebrafish embryo. It would be fruitful in future work to assess the cost-precision trade-off of more comprehensive models by incorporating the thermodynamic costs of other components of the signaling network underlying biological pattern formation. These discussion points are added throughout the revised manuscript.

– Rodenfels, J., Neugebauer, K. M., and Howard, J. (2019). Heat Oscillations Driven by the Embryonic Cell Cycle Reveal the Energetic Costs of Signaling. *Dev. Cell*, 48(5):646–658.

– Rodenfels, J., Sartori, P., Golfier, S., Nagendra, K., Neugebauer, K., and Howard, J. (2020). Contribution of increasing plasma membrane to the energetic cost of early zebrafish embryogenesis. *MBoC*, 31(7):520–526.

– Song, Y., Park, J. O., Tanner, L., Nagano, Y., Rabinowitz, J. D., and Shvartsman, S. Y. (2019). Energy budget of *Drosophila* embryogenesis. *Curr. Biol.*, 29(12):R566–R567

Our response to the choice of trade-off function and its robustness is given in 2. (please see below).

As mentioned above, the revised manuscript contains a more detailed discussion on the cost of generating the morphogen profile. The proportionality constant, αo, in the expression of the cost C≡αo(vcelljin/lcell)τ (Equation 5) represents the thermodynamic cost (free energy) involving synthesizing and degrading a single morphogen molecule, which can be quantified by the number of ATPs hydrolyzed in the process. For instance, the thermodynamic cost required for the synthesis and degradation of a single Bcd protein composed of 494 amino acids is estimated to be on the order of αo≈494×4≈2×103 ATPs, where we assume that each peptide-bond formation requires 4 ATPs (Lynch et al., 2015). During the first ~2 hours of *Drosophila* embryogenesis when the anterior-posterior axis patterning takes place, about ~5×108 Bcd molecules are produced (Drocco et al., 2012). Thus, the cost of generating the Bcd profile is on the order of 10^12^ ATPs, which is only a small fraction of the total energy budget of *Drosophila* embryogenesis ~7×1016 ATPs (Song et al., 2019).

However, the energetic cost of numerous individual molecular processes comprise the overall energy budget of an organism. Thus, it is of great interest to quantify the energetic efficiency of each molecular process. For instance, molecular motors (e.g. kinesin) and biomass-producing enzymes (e.g. ribosome) balance multiple functional features, such as the thermodynamic cost, reaction speed, and the fluctuations of the reactions. In light of the thermodynamic uncertainty relation, the product Q≡ΔS(t)⟨δX(t)2⟩⟨X(t)⟩2 quantifies the trade-offs among the thermodynamic cost (ΔS(t)), speed (⟨X(t)⟩2), and the fluctuations (⟨δX(t)2⟩) of the time-dependent observable X(t), which can be defined as the steps taken by the molecular motor or the number of peptide-bonds synthesized by the ribosome. Previous works by Hyeon and colleagues have shown that Q of certain molecular processes are close to the physical lower bound of 2kB (Q≈7−15kB for the molecular motors, and Q≈45−50kB for *E. coli* ribosome), which suggests that those molecular processes have evolved to operate cost-effectively (Hwang and Hyeon, *JPCL* 2018; Song and Hyeon, *JPCL* 2020; Song and Hyeon, *JCP* 2021). Similarly, in our study, it is remarkable to find that the fruit fly embryos are nearly optimized with respect to the cost-precision trade-off product (π), which suggests that both the cost and the precision are key physical features that determine the shape of morphogen profiles at the time scale of evolution.

While it is possible to assign explicit values to the cost of generating the morphogen profile, C, the interpretation of our theoretical result in the context of the biological system must be done with care. For the Bcd profile with the boundary position xb≈0.4L, πo(xb,λBcd)≈αo(L/lcell)0.6 (Figure 2b) where L/lcell≈50, and the experimentally reported precision is ϵ2(xb)≈7×10−4 (Gregor et al., 2007). Thus, the cost associated with the Bcd profile is C(xb,λBcd)=πo(xb,λBcd)/ϵ2(xb,λBcd)≈40000αo, or equivalently, the thermodynamic cost of synthesizing and degrading 40000 Bcd molecules. This is orders of magnitude smaller than the estimate of 5×108 based on the direct measurement of the total Bcd synthesis in the embryo (Drocco et al., 2012). The discrepancy between the two numbers most likely arises because our theory simplifies the dynamics of the nuclear concentration of Bcd in one dimension, whereas the direct measurement of Bcd synthesis accounts for all the molecules in the 3D volume including the cytoplasm and the nuclei. Additionally, the duration to generate the steady state morphogen profile is longer than the characteristic time, τ. However, assuming that the cytoplasmic and nuclear concentrations are proportional to each other, and that the time for the morphogen profile to reach steady states is proportional to τ, it is possible to adjust the proportionality constant αo so that C(xb) represents the overall cost to produce the concentration profile of nuclear Bcd.

– Lynch, M. and Marinov, G. K. (2015). The bioenergetic costs of a gene. *Proc. Natl. Acad. Sci. USA,* 112(51):15690– 15695.

– Gregor, T., Tank, D. W., Wieschaus, E. F., and Bialek, W. (2007). Probing the Limits to Positional Information. *Cell*, 130(1):153–164.

– Drocco, J. A., Wieschaus, E. F., and Tank, D. W. (2012). The synthesis-diffusion-degradation model explains Bicoid gradient formation in unfertilized eggs. *Phys. Biol.*, 9(5):055004.

– Hwang, W. and Hyeon, C. (2018). Energetic Costs, Precision, and Transport Efficiency of Molecular Motors. *J. Phys. Chem. Lett.*, 9(3):513–520.

– Song, Y. and Hyeon, C. (2020). Thermodynamic cost, speed, fluctuations, and error reduction of biological copy machines. *J. Phys. Chem. Lett.*, 11:3136–3143.

– Song, Y. and Hyeon, C. (2021). Thermodynamic uncertainty relation to assess the efficiency of biological processes. *J. Chem. Phys.*, 154:130901.

2) Similarly, the authors choose a product of thermodynamic cost and the squared relative error in the position as a trade-off function that they analyze. My question is how robust are the obtained results? The choice of the trade-off function is not unique and it does not have any fundamental foundation, to the best of my understanding. For instance, one defines X as a thermodynamic cost and Y as relative error in position, then not only X*Y, but also (X+Y), X*Y*Y, X*X*Y and many other combinations also might be viewed as trade-off functions. But they would lead to a different prediction on the optimized minimal length and other related issues. Then how to choose and why?

The main reason for defining the trade-off by π≡C×ϵ2 was the following:

There are two quantities of interest in the problem of cost-precision trade-off of morphogen profile formation. First, researchers interested in the precision of morphogen concentration at the target boundary have been measuring the quantity which amounts to “the relative error of morphogen concentration” defined to be proportional to the ratio of concentration fluctuation to its mean value, namely, σρ/⟨ρ⟩. In fact, this quantity, often used in statistics, is proportional to 1/nm1/2, which is nothing but the outcome of central limit theorem. Second, the cost of producing morphogens (C), which is the main concern in our study, is expected to be proportional to the number of morphogens being produced (nm). – Let us clarify here that the cost (C) we refer to in our study is specific to the morphogen, not the whole cellular budget. – Then, when the cost of morphogen production is large, the precision of morphogen profile at the target boundary is high (or the relative error of morphogen concentration at the target boundary is small). Conversely, when the cost is small, the precision is low (or the error is large). This means that there is a trade-off between the cost and precision. The simplest and generic construction for discussing the trade-off relationship between the two quantities, independently from nm (i.e. the system size) is to consider the product between the cost (C) and the squared value of σρ/⟨ρ⟩.

Let us emphasize again that π=C×ϵ2 , which we called the trade-off product, is a dimensionless quantity, independent of the number of morphogens being produced (nm).

Other constructions suggested by the referee(s), such as X+Y, X*Y*Y, and X*X*Y do not fulfill this requirement, and the minimization of these constructions leads to physically uninterpretable conclusions.

In fact, this form of the cost-precision trade-off was motivated by the thermodynamic uncertainty relation (TUR), which defines the product between the squared relative error of the time integrated current-like observable X(t) and the total entropy production for t, ΔS(t), namely, Q≡ΔS(t)⟨δX(t)2⟩⟨X(t)⟩2. In the case of TUR, the two quantities of interest are a function of time *t*, such that ΔS(t)∼t, ⟨δX(t)2⟩⟨X(t)⟩2∼1/t, and the dependence of time *t* is cancelled off when the product of the two is taken. What is remarkable in TUR is that there exists a model independent universal bound in the product, Q≥2kB, where kB is the Boltzmann constant, and the cost-effectiveness of a dynamical process generated in nonequilibrium is dictated by how close the parameter *Q* is to its minimal bound.

Regarding the robustness of our result, we would like to emphasize this: For the scenarios of the point measurement and the space-time averaging, we addressed the cost-precision trade-off of the morphogen gradient formation by employing slightly different definitions of the cost and the precision. We obtained similar conclusions as to the optimal characteristic lengths, λmin≈0.43xb for the point measurement and λmin≈0.5xb for the space-time-averaged measurement, both of which are the outcomes of minimizing the respective trade-off product.

3) The work constraints the patterning problem to a gradient which produces a single target boundary, but validates the results with measurements in systems where gradients are known to regulate many downstream patterns. The result of the trade-off analysis is that the location of the target boundary determines the shape of the gradient. How is this reconciled with the understanding that, for example, the Bicoid gradient is read out by multiple downstream genes along the entire length of the fly embryo? (Hannon et al., eLife, 2017).

Our theory implies that the precision of the morphogen induced spatial boundary can be optimized cost-effectively at only one spatial position. For the morphogens Bcd, Dpp, and Wg, our analysis focused on the target gene expressions whose quantitative characterization revealed a relatively sharp boundary (Figure 2-S1). In the case of Hh, we included the full range of spatial boundaries formed by the three target genes, *dpp*, *col*, and *en* (Figure 2-S1). As the referees point out, however, it is conceivable that the precision of the morphogen profile is essential at multiple boundary positions. One can consider an extension of our study in which a weighted average of the precision of many boundary positions is balanced against the total cost required to form the profile. This discussion is added to the revised manuscript.

4) The description of the error above Equation 1 is somewhat confusing in two different ways: i) definitions are given in terms of mathematical variables like N and the subscript i that are not explicitly explained. The reader can figure out the intended meaning via later discussions, but it would be helpful to make these terms clear from the start. ii) More broadly, the notion of "measurement" used in the paper is a little abstract, and is likely to confuse biologically-oriented readers. Before delving into the mathematical definition of measurement errors, could the authors provide a summary of how measurements could hypothetically be carried out via concrete biological mechanisms? For example, they mention that space-time-averaged measurements might be carried out by cell surface receptors. But what mechanisms could lead to instantaneous point measurements?

Thank you for this suggestion. In the revised manuscript, we clarified that Bcd concentration at each nucleus can be detected in terms of the frequency of Bcd binding to regulatory sequences in DNA. Higher local concentration of Bcd leads to a more frequent expression of the target gene, resulting in its position-dependent expression profile. Additionally, cells with many receptors positioned across the cell surface may perform space-time-averaging by integrating the signal from morphogen-bound receptors for an extended time interval. Theoretically, cells may control the duration of the measurement by modulating the availability of morphogen-sensing receptors, or by tuning the overall transcription rate of the target gene. Then, the point measurement is effectively interpreted as the short time limit of the space-time averaged counterpart with a single receptor. Alternatively, one can define the point measurement of the morphogen profile as an image of the morphogen profile obtained from a fixed sample.

5) In line 181 the authors state that "the two limiting models can be effectively minimized by similar λ values at a given position is of great significance." However comparing Figures 2(c) and 3(c) it is a hard to directly see this, because in one case the y-axis is normalized as λ/L and in the other case as λ/a. Could the authors somehow make this similarity in predicted λ values clearer? Maybe adding a twin y axis scale to one of the figures, showing λ in terms of the other normalization?

Our main finding can be summarized by the expression of the optimal characteristic length λ that minimizes the trade-off product at a given target boundary. In Figure 3c of the revised manuscript, we have overlaid the linear approximation of the optimal characteristic length λmin from the point measurement model with λmin from the model with space-time-averaging, so that the readership can compare the two results side-by-side. Additionally, in the discussion, we clarified that λmin of the two models were similar, in that λmin≈0.43L in the former and λmin≈0.5L in the latter.

6) The discussion of the reversible model in the appendix is quite nice, particularly how you can derive all the irreversible results directly from the reversible model by taking an appropriate limit. Particularly interesting is that the cost defined in terms of entropy production rate times characteristic time, if normalized by ln(γ) [using Equation 33] would seem to give you the morphogen production cost of Equation 37, up to a factor of α_0_, assuming the irreversible limit where ln(γ) -> infinity. So it seems that the entropy-based cost is closely related to the other definition of cost, and the authors could consider highlighting this fact in the main text. Is there any qualitative way of understanding why ln(γ) is the appropriate normalization factor?

We appreciate your suggestion. The reversible model motivates a physical interpretation of the proportionality constant, αo, which we defined in the irreversible model of morphogen dynamics as the number of ATPs hydrolyzed for the synthesis and degradation of a morphogen molecule. When the entropy production rate (S˙tot, Equation 32) is normalized by ln(γ), and vout and rs are negligibly small, we obtain the equality limrs,vout→0S˙tot/lnγ=vcelljin/lcell (Equation 33). Comparison of this with the expression of C (Equation 5) enables us to relate αo with the free energy cost, Tln(γ). Thus, the thermodynamic cost associated with the morphogen profile is expressed as C≡(αovcelljin/lcell)τ (Equation 5) where τ is the time required to generate the profile and αovcelljin/lcell approximates the rate of total dissipation, TS˙tot. In the revised manuscript, we direct the readership to the Appendix, where this discussion has been added.

7) All derivations are mode for the finite length L. However, as argued in the specific calculations L>>λ, so why not to derive the results for L>>1. In this case, all the formulas are much simpler and there is a clear physical interpretation for all the quantities. Why the bulkier and less transparent expressions are used in the text? If authors wanted to explain things better to a broader readership and to emphasize the physics they better use a simple analytical framework. Or they need to explain why they used a finite L.

The revised manuscript includes the more transparent expressions at the limit of large L.

8) It would be better to describe fully how the characteristic lengths were estimated. This is because the reported numbers do not fully agree with what is reported in experiments. For example, for BCD Berezhkovskii and Shvartsman report λ=75 nm, not 100nm as given in this paper. Why are such different results are reported?

In the plots, we used the characteristic lengths *reported from each experimental study*. In the case of the Bcd profile, there have been slight differences depending on the reference that reports the characteristic length. However, relatively small discrepancies of the characteristic lengths do not change the qualitative conclusions of our work. For more clarity, we added guiding lines in Figure 2-S1 with the experimental measurements of the morphogen profiles, which show that the approximations of exponentially decaying profiles with the reported characteristic lengths fits the raw data faithfully.